# A Bandit Framework for Strategic Regression

**Yang Liu and Yiling Chen**
School of Engineering and Applied Science, Harvard University
{yangl,yiling}@seas.harvard.edu

## Abstract

We consider a learner's problem of acquiring data dynamically for training a regression model, where the training data are collected from strategic data sources. A fundamental challenge is to incentivize data holders to exert effort to improve the quality of their reported data, despite that the quality is not directly verifiable by the learner. In this work, we study a dynamic data acquisition process where data holders can contribute multiple times. Using a bandit framework, we leverage the long-term incentive of future job opportunities to incentivize high-quality contributions. We propose a Strategic Regression-Upper Confidence Bound (SR-UCB) framework, a UCB-style index combined with a simple payment rule, where the index of a worker approximates the quality of his past contributions and is used by the learner to determine whether the worker receives future work. For linear regression and a certain family of non-linear regression problems, we show that SR-UCB enables an $O\left(\sqrt{\log T/T}\right)$-Bayesian Nash Equilibrium (BNE) where each worker exerts a target effort level that the learner has chosen, with $T$ being the number of data acquisition stages. The SR-UCB framework also has some other desirable properties: (1) The indexes can be updated in an online fashion (hence computation is light). (2) A slight variant, namely Private SR-UCB (PSR-UCB), is able to preserve $(O\left(\log^{-1} T\right), O\left(\log^{-1} T\right))$-differential privacy for workers' data, with only a small compromise on incentives (each worker exerting a target effort level is an $O\left(\log^6 T/\sqrt{T}\right)$-BNE).

## 1 Introduction

More and more data for machine learning nowadays are acquired from distributed, unmonitored and strategic data sources and the quality of these collected data is often unverifiable. For example, in a crowdsourcing market, a data requester can pay crowd workers to label samples. While this approach has been widely adopted, crowdsourced labels have been shown to degrade the learning performance significantly, see e.g., [19], due to the low quality of the data. How to incentivize workers to contribute high-quality data is hence a fundamental question that is crucial to the long-term viability of this approach.

Recent works [2,4,10] have considered incentivizing data contributions for the purpose of estimating a regression model. For example Cai et al. [2] design payment rules so that workers are incentivized to exert effort to improve the quality of their contributed data, while Cummings et al. [4] design mechanisms to compensate privacy-sensitive workers for their privacy loss when contributing their data. These studies focus on a static data acquisition process, only considering one-time data acquisition from each worker. Hence, the incentives completely rely on the payment rule. However, in stable crowdsourcing markets, workers return to receive additional work. Future job opportunities are thus another dimension of incentives that can be leveraged to motive high-quality data contributions. In this paper, we study dynamic data acquisition from strategic agents for regression problems and explore the use of future job opportunities to incentivize effort exertion.

In our setting, a learner has access to a pool of workers and in each round decides on which workers to ask for data. We propose a Multi-armed Bandit (MAB) framework, called Strategic Regression-Upper Confidence Bound (SR-UCB), that combines a UCB-style index rule with a simple per-round payment rule to align the incentives of data acquisition with the learning objective. Intuitively, each worker is an arm and has an index associated with him that measures the quality of his past contributions. The indexes are used by the learner to select workers in the next round. While MAB framework is natural for modeling selection problem with data contributors of potentially varying qualities, our setting has two challenges that are distinct from classical bandit settings. First, after a worker contributes his data, there is no ground-truth observation to evaluate how well the worker performs (or reward as commonly referred to in a MAB setting). Second, a worker's performance is a result of his strategic decision (e.g. how much effort he exerts), instead of being purely exogenously determined. Our SR-UCB framework overcomes the first challenge by evaluating the quality of an agent's contributed data against an estimator trained on data provided by all other agents to obtain an unbiased estimate of the quality, an idea inspired by the peer prediction literature [11, 16]. To address the second challenge, our SR-UCB framework enables a game-theoretic equilibrium with workers exerting target effort levels chosen by the learner. More specifically, in addition to proposing the SR-UCB framework, our contributions include:

- We show that SR-UCB helps simplify the design of payment, and successfully incentivizes effort exertion for acquiring data for linear regression. Every worker exerting a targeted effort level (for labeling and reporting the data) is an $O\left(\sqrt{\log T/T}\right)$-Bayesian Nash Equilibrium (BNE). We can also extend the above results to a certain family of non-linear regression problems.
- SR-UCB indexes can be maintained in an online fashion, hence are computationally light.
- We extend SR-UCB to Private SR-UCB (PSR-UCB) to further provide privacy guarantees, with small compromise on incentives. PSR-UCB is $(O\left(\log^{-1} T\right), O\left(\log^{-1} T\right))$-differentially private and every worker exerting the targeted effort level is an $O\left(\log^6 T/\sqrt{T}\right)$-BNE.

## 2   Related work

Recent works have formulated various strategic learning settings under different objectives [2, 4, 10, 20]. Among these, payment based solutions are proposed for regression problems when data come from workers who are either effort sensitive [2] or privacy sensitive [4]. These solutions induce game-theoretic equilibria where high-quality data are contributed. The basic idea of designing the payment rules is inspired by the much mature literature of proper scoring rules [8] and peer prediction [16]. Both [2] and [4] consider a static data acquisition procedure, while our work focuses on a dynamic data acquisition process. Leveraging the long-term incentive of future job opportunities, our work has a much simpler payment rule than those of [2] and [4] and relaxes some of the restrictions on the learning objectives (e.g., well behaved in [2]), at the cost of a weaker equilibrium concept (approximate BNE in this work vs. dominate strategy in [2]).

Multi-armed Bandit (MAB) is a sequential decision making and learning framework which has been extensively studied. It is nearly impossible to survey the entire bandit literature. The seminal work by Lai et al [13] derived lower and upper bounds on asymptotic regret on bandit selection. More recently, finite-time algorithms have been developed for i.i.d. bandits [1] . Different from the classical settings, this work needs to deal with challenges such as no ground-truth observations for bandits and bandits' rewards being strategically determined. A few recent works [7, 15] also considered bandit settings with strategic arms. Our work differs from these in that we consider a regression learning setting without ground-truth observations, as well as we consider long-term workers whose decisions on reporting data can change over time.

Our work and motivations have some resemblance to online contract design problems for a principal-agent model [9]. But unlike the online contract design problems, our learner cannot verify the quality of finished work after each task assignment. In addition, instead of focusing on learning the optimal contract, we use bandits mainly to maintain a long-term incentive for inducing high-quality data.

## 3   Formulation

The learner observes a set of feature data $X$ for training. To make our analysis tractable, we assume each $x \in X$ is sampled uniformly from a unit ball with dimension $d$: $x \in \mathbb{R}^d$ s.t. $||x||_2 \le 1$. Each

$x$ associates with a ground-truth response (or label) $y(x)$, which cannot be observed directly by the learner. Suppose $x$ and $y(x)$ are related through a function $f : \mathbb{R}^d \to \mathbb{R}$ that $y(x) = f(x) + z$, where $z$ is a zero-mean noise with variance $\sigma_z$, and is independent of $x$. For example, for linear regression $f(x) = \theta^T x$ for some $\theta \in \mathbb{R}^d$. The learner would like to learn a good estimate $\tilde{f}$ of $f$. For the purpose of training, the learner needs to figure out $y(x)$ for different $x \in X$. To obtain an estimate $\tilde{y}(x)$ of $y(x)$, the learner assigns each $x$ to a selected worker to obtain a label.

**Agent model:** Suppose we have a set of workers $\mathcal{U} = \{1, 2, ..., N\}$ with $N \geq 2$. After receiving the labeling task, each worker will decide on the effort level $e$ he wants to exert to generate an outcome – higher effort leads to a better outcome, but is also associated with a higher cost. We assume $e$ has bounded support $[0, \bar{e}]$ for all worker $i \in \mathcal{U}$. When deciding on an effort level, a worker wants to maximize his expected payment minus cost for effort exertion. The resulted label $\tilde{y}(x)$ will be given back to the learner. Denote by $\tilde{y}_i(x, e)$ the label returned by worker $i$ for data instance $x$ (if assigned) with chosen effort level $e$. We consider the following effort-sensitive agent model: $\tilde{y}_i(x, e) = f(x) + z + z_i(e)$, where $z_i(e)$ is a zero-mean noise with variance $\sigma_i(e)$. $\sigma_i(e)$ can be different for different workers, and $\sigma_i(e)$ decreases in $e, \forall i$. The $z$ and $z_i$'s have bounded support such that $|z|, |z_i| \leq Z, \forall i$. Without loss of generality, we assume that the cost for exerting effort $e$ is simply $e$ for every worker.

**Learner's objective** Suppose the learner wants to learn $f$ with the set of samples $X$. Then the learner finds effort levels $\mathbf{e}^*$ for data points in $X$ such that

$$\mathbf{e}^* \in \mathrm{argmin}_{\{e(x)\}_{x \in X}} \mathrm{ERROR}(\tilde{f}(\{x, \tilde{y}(x, e(x))\}_{x \in X})) + \lambda \cdot \mathrm{PAYMENT}(\{e(x)\}_{x \in X}) \,,$$

where $e(x)$ is the effort level for sample $x$, and $\{\tilde{y}(x, e(x))\}_{x \in X}$ is the set of labeled responses for training data $X$. $\tilde{f}(\cdot)$ is the regression model trained over this data. The learner assigns the data and pay appropriately to induce the corresponding effort level $\mathbf{e}^*$. This formulation resembles the one presented in [2]. The ERROR term captures the expected error of the trained model using collected data (e.g., measure in squared loss), while the PAYMENT term captures the total expected budget that the learner spends to receive the labels. This payment quantity depends on the mechanism that the learner chooses to use and is the expected payment of the mechanism to induce selected effort level for each data point $\{e(x)\}_{x \in X}$. $\lambda > 0$ is a weighting factor, which is a constant. It is clear that the objective function depends on $\sigma_i$'s. We assume for now that the learner knows $\sigma_i(\cdot)$'s,[1] and the optimal $\mathbf{e}^*$ can be computed.

## 4 StrategicRegression-UCB (SR-UCB): A general template

We propose SR-UCB for solving the dynamic data acquisition problem. SR-UCB enjoys a bandit setting, where we borrow the idea from the classical UCB algorithm [1], which maintains an index for each arm (worker in our setting), balancing exploration and exploitation. While a bandit framework is not necessarily the best solution for our dynamic data acquisition problem, it is a promising option for the following reasons. First, as utility maximizers, workers would like to be assigned tasks as long as the marginal gain for taking a task is positive. A bandit algorithm can help execute the assignment process. Second, carefully designed indexes can potentially reflect the amount of effort exerted by the agents. Third, because the arm selection (of bandit algorithms) is based on the indexes of workers, it introduces competition among workers for improving their indexes.

SR-UCB contains the following two critical components:

**Per-round payment** For each worker $i$, once selected to label a sample $x$, we will assign a base payment $p_i = e_i + \gamma$,[2] after reporting the labeling outcome, where $e_i$ is the desired effort level that we would like to induce from worker $i$ (for simplicity we have assumed the cost for exerting effort $e_i$ equals to the effort level), and $\gamma > 0$ is a small quantity. The design of this base payment is to ensure once selected, a worker's base cost will be covered. Note the above payment depends on neither the assigned data instance $x$ nor the reported outcome $\tilde{y}$. Therefore such a payment procedure can be pre-defined after the learner sets a target effort level.

**Assignment** The learner assigns multiple task $\{x_i(t)\}_{i \in d(t)}$ at time $t$, with $d(t)$ denoting the set of workers selected at $t$. Denote by $e_i(t)$ the effort level worker $i$ exerted for $x_i(t)$, if $i \in d(t)$. Note all $\{x_i(t)\}_{i \in d(t)}$ are different tasks, and each of them is assigned to exactly one worker. The selection of workers will depend on the notion of indexes. Details are given in Algorithm 1.

---

**Algorithm 1** SR-UCB: Worker index & selection

---

**Step 1**. For each worker $i$, first train estimator $\tilde{f}_{-i,t}$ using data $\{x_j(n) : 1 \leq n \leq t-1, \ j \in d(n), j \neq i\}$, that is using the data collected from workers $j \neq i$ up to time $t-1$. When $t = 1$, we will initialize by sampling each worker at least once such that $\tilde{f}_{-i,t}$ can be computed.

**Step 2**. Then compute the following index for worker $i$ at time $t$

$$I_i(t) = \frac{1}{n_i(t)} \sum_{n=1}^{t} 1(i \in d(n)) \left[ a - b \left( \tilde{f}_{-i,t}(x_i(n)) - \tilde{y}_i(n, e_i(n)) \right)^2 \right] + c \sqrt{\frac{\log t}{n_i(t)}} \, ,$$

where $n_i(t)$ is the number of times worker $i$ has been selected up to time $t$. $a, b$ are two positive constants for "scoring", and $c$ is a normalization constant. $\tilde{y}_i(n, e_i(n))$ is the corresponding label for task $x_i(n)$ with effort level $e_i(n)$, if $i \in d(n)$.

**Step 3**. Based on the above index, we select $d(t)$ at time $t$ such that $d(t) := \{ j : I_j(t) \geq \max_i I_i(t) - \tau(t) \}$, where $\tau(t)$ is a perturbation term decreasing in $t$.

---

Some remarks on SR-UCB: (1) Different from the classical bandit setting, when calculating the indexes, there is no ground-truth observation for evaluating the performance of each worker. Therefore we adopt the notion of scoring rule [8]. Particularly the one we used above is the well-known Brier scoring rule: $B(p,q) = a - b(p-q)^2$ . (2) The scoring rule based index looks similar to the payment rules studied in [2, 4]. But as we will show later, under our framework the selection of $a, b$ is much less sensitive to different problem settings, as with an index policy, only the relative values matter (ranking). This is another benefit of separating payment from selection. (3) Instead of only selecting the best worker with the highest index, we select workers whose index is within a certain range of the maximum one (a confidence region). This is because workers may have competing expertise level and hence selecting only one of them would de-incentivize workers' effort exertion.

### 4.1 Solution concept

Denote by $\mathbf{e}(n) := \{e_1(n), ..., e_N(n)\}$, and $e_{-i}(n) = \{e_j(n)\}_{j \neq i}$. We define approximate Bayesian Nash Equilibrium as our solution concept:

**Definition 1.** Suppose SR-UCB runs for $T$ stages. $\{e_i(t)\}_{i=1,t=1}^{N,T}$ is a $\pi$-BNE if $\forall i, \{\tilde{e}_i(t)\}_{t=1}^{T}$:

$$\frac{1}{T} \mathbb{E}[\sum_{t=1}^{T} (p_i - e_i(t)) 1(i \in d(t)) | \{\mathbf{e}(n)\}_{n \leq t}] \geq \frac{1}{T} \mathbb{E}[\sum_{t=1}^{T} (p_i - \tilde{e}_i(t)) 1(i \in d(t)) | \{\tilde{e}_i(n), e_{-i}(n)\}_{n \leq t}] - \pi.$$

This is to say by deviating, each worker will gain no more than $\pi$ net-payment per around. We will establish our main results in terms of $\pi$-BNE. The reason we adopt such a notion is that in a sequential setting it is generally hard to achieve strict BNE or other stronger notion as any one-step deviation may not affect a long-term evaluation by much.[3] Approximate BNE is likely the best solution concept we can hope for.

## 5 Linear regression

### 5.1 Settings and a warm-up scenario

In this section we present our results for a simple linear regression task where the feature $x$ and observation $y$ are linearly related via an unknown $\theta$: $y(x) = \theta^T x + z$, $\forall x \in X$. Let's start with assuming all workers are statistically identical such that $\sigma_1 = \sigma_2 = ... = \sigma_N$. This is an easier case that serves as a warm-up. It is known that given training data, we can find an estimation $\tilde{\theta}$ that minimizes a

non-regularized empirical risk function: $\tilde{\theta} = \text{argmin}_{\hat{\theta} \in \mathbb{R}^d} \sum_{x \in X} (y(x) - \hat{\theta}^T x)^2$ (linear least square). To put this model into SR-UCB, denote $\tilde{\theta}_{-i}(t)$ as the linear least square estimator trained using data from workers $j \neq i$ up to time $t-1$. And $I_i(t) := S_i(t) + c\sqrt{\log t / n_i(t)}$, with

$$S_i(t) := \frac{1}{n_i(t)} \sum_{n=1}^{t-1} \mathbb{1}(i \in d(n)) \left[ a - b \left( \tilde{\theta}_{-i}^T(t) x_i(n) - \tilde{y}_i(n, e_i(n)) \right)^2 \right]. \qquad (5.1)$$

Suppose $||\theta||_2 \leq M$. Given $||x||_2 \leq 1$ and $|z|, |z_i| \leq Z$, we then prove that $\forall t, n, i$, $(\tilde{\theta}_{-i}^T(t) x_i(n) - \tilde{y}_i(n, e_i(n)))^2 \leq 8M^2 + 2Z^2$. Choose $a, b$ such that $a - (8M^2 + 2Z^2)b \geq 0$, then we have $0 \leq S_i(t) \leq a$, $\forall i, t$. For the perturbation term, we set $\tau(t) := O(\sqrt{\log t / t})$. The intuition is that with $t$ samples, the uncertainties in the indexes, coming from both the score calculation and the bias term, can be upper bounded at the order of $O(\sqrt{\log t / t})$. Thus, to not miss a competitive worker, we set the tolerance to be at the same order.

We now develop the formal equilibrium result of SR-UCB for linear least square. Our analysis requires the following assumption on the smoothness of $\sigma$.

**Assumption 1.** *We assume $\sigma(e)$ is convex on $e \in [0, \bar{e}]$, with gradient $\sigma'(e)$ being both upper bounded, and lower bounded away from 0, i.e., $\overline{L} \geq |\sigma'(e)| \geq \underline{L} > 0$, $\forall e$.*

The learner wants to learn $f$ with a total of $NT$ $(= |X|$ or $\lceil NT \rceil = |X|)$ samples. Since workers are statistically equivalent, ideally the learner would like to run SR-UCB for $T$ steps and collect a label for a unique sample from each worker at each step. Hence, the learner would like to elicit a single target effort level $e^*$ from all workers and for all samples:

$$e^* \in \text{argmin}_e \mathbb{E}_{x,y,\tilde{y}} \left[ \theta^T (\{x_i(n), \tilde{y}_i(n, e)\}_{i=1,n=1}^{N,T}) \cdot x - y \right]^2 + \lambda \cdot (e + \gamma) NT. \qquad (5.2)$$

Due to the uncertainty in worker selection, it is highly likely that after step $T$, there will be tasks left unlabelled. We can let the mechanism go for extra steps to complete labelling of these tasks. But due to the bounded number of missed selections as we will show later, stopping at step $T$ won't affect the accuracy in the model trained.

**Theorem 1.** *Under SR-UCB for linear least square, set fixed payment $p_i = e^* + \gamma$ for all $i$, where $\gamma = \Omega(\sqrt{\log T / T})$, choose $c$ to be a large enough constant, $c \geq Const.(M, Z, N, b)$, and let $\tau(t) := O(\sqrt{\log t / t})$. Workers have full knowledge of the mechanism and the values of the parameters. Then at an $O(\sqrt{\log T / T})$-BNE, workers, whenever selected, exert effort $e_i(t) \equiv e^*$ for all $i$ and $t$.*

The net payment (payment minus the cost of effort) per task can be made arbitrarily small by setting $\gamma$ exactly on the order of $O(\sqrt{\log T / T})$, and $p_i - e^* = \gamma = O(\sqrt{\log T / T}) \to 0$, as $T \to \infty$.

Our solution heavily relies on forming a race among workers. By establishing the convergence of bandit indexes to a function of effort (via $\sigma(\cdot)$), we show that when other workers $j \neq i$ follow the equilibrium strategy, worker $i$ will be selected w.h.p. at each round, if he also puts in the same amount of effort. On the other hand, if worker $i$ shirks from doing so by as much as $(O(\sqrt{\log T / T}))$, his number of selection will go down in order. This establishes the $\pi$-BNE. As long as there exists one competitive worker, all others will be incentivized to exert effort. Though as will be shown in the next section, all workers shirking from exerting effort is also an $O(\sqrt{\log T / T})$-BNE. This equilibrium can be removed by adding some uncertainty on top of the bandit selection procedure. When there are $\geq 2$ workers being selected in SR-UCB, each of them will be assigned a task with certain probability $0 < p_s < 1$. While when there is a single selected worker, the worker is assigned a task w.p. 1. Set $p_s := 1 - O(\sqrt{\log T / T}/\gamma)$. So with probability $1 - p_s = O(\sqrt{\log T / T}/\gamma)$, even the "winning" workers will miss the selection. With this change, exerting $e^*$ still forms an $O(\sqrt{\log T / T})$-BNE, while every worker exerting any effort level that is $\Delta e > O(\gamma)$ lower than the target effort level is not a $\pi$-BNE with $\pi \leq O(\sqrt{\log T / T})$.

## 5.2 Linear regression with different σ

Now we consider the more realistic case that different workers have different noise-effort function $\sigma$'s. W.l.o.g., we assume $\sigma_1(e) < \sigma_2(e) < ... < \sigma_N(e), \forall e$.[4] In such a setting, ideally we would always like to collect data from worker 1 since he has the best expertise level (lowest variance in labeling noise). Suppose we are targeting an effort level $e_1^*$ from data source 1 (the best data source). We first argue that we also need to incentivize worker 2 to exert competitive effort level $e_2^*$ such that $\sigma_1(e_1^*) = \sigma_2(e_2^*)$, and we assume such an $e_2^*$ exists.[5] This also naturally implies that $e_2^* > e_1^*$ as worker 1 contributes data with less variance in noise at the same effort level. The reason is similar to the homogeneous setting—over time workers form a competition on $\sigma_i(e_i)$. Having a competitive peer will motivate workers to exert as much effort as he can (up to the payment). Therefore the goal for such a learner (with $2T$ samples to assign) is to find an effort level $e^*$ such that [6]

$$e^* \in \text{argmin}_{e_2:\sigma_1(e_1)=\sigma_2(e_2)} \mathbb{E}_{x,y,\tilde{y}} \left[ \theta^T(\{x_i(n), \tilde{y}_i(n,e_i))\}_{i=1,n=1}^{2,T})x - y \right]^2 + \lambda \cdot (e_2 + \gamma)2T.$$

Set the one-step payment to be $p_i = e^* + \gamma, \forall i$. Let $e_1^*$ be the solution to $\sigma_1(e_1^*) = \sigma_2(e^*)$ and let $e_i^* = e^*$ for $i \geq 2$. Note for $i > 2$ we have $\sigma_i(e_i^*) - \sigma_1(e_1^*) > 0$. While we have argued about the necessity for choosing the top two most competitive workers, we have not mentioned the optimality of doing so. In fact selecting the top two is the best we can do. Suppose on the contrary, the optimal solution is by selecting top $k > 2$ workers, at effort level $e_k$. According to our solution, we targeted the effort level that leads to variance of noise $\sigma_k(e_k)$ (so the least competitive worker will be incentivized). Then we can simply target the same effort level $e_k$, but migrating the task loads to only the top two workers – this keeps the payment the same, but the variance of noise now becomes $\sigma_2(e_k) < \sigma_k(e_k)$, which leads to better performance. Denote $\Delta_1 := \sigma_3(e^*) - \sigma_1(e_1^*) > 0$ and assume Assumption 1 applies to all $\sigma_i$'s. We prove:

**Theorem 2.** *Under SR-UCB for linear least square, set $c \geq Const.(M, Z, b, \Delta_1)$, $\Omega(\sqrt{\log T / T}) = \gamma \leq \frac{\Delta_1}{2L}$, $\tau(t) := O(\sqrt{\log t / t})$. Then, each worker $i$ exerting effort $e_i^*$ once selected forms an $O(\sqrt{\log T / T})$-BNE.*

**Performance with acquired data** If workers follow the $\pi$-BNE, the contributed data from the top two workers (who have been selected the most number of times) will have the same variance $\sigma_1(e_1^*)$. Then following results in [4], w.h.p. the performance of the trained classifier is bounded by $O(\sigma_1(e_1^*)/(\sum_{i=1,2} n_i(T))^2)$. Ideally we want to have $\sum_{i=1,2} n_i(T) = 2T$, such that an upper bound of $O(\sigma_1(e_1^*)/(2T)^2)$ can be achieved. Compared to the bound $O(\sigma_1(e_1^*)/(2T)^2)$, SR-UCB's expected performance loss (due to missed sampling & wrong selection, which is bounded at the order of $O(\log T)$) is bounded by $\mathbb{E}[\sigma_1(e_1^*)/(\sum_{i=1,2} n_i(T))^2 - \sigma_1(e_1^*)/(2T)^2] \leq O(\sigma_1(e_1^*)\log T/T^3)$ w.h.p. .

**Regularized linear regression** Ridge estimator has been widely adopted for solving linear regression. The objective is to find a linear model $\tilde{\theta}$ that minimizes the following regularized empirical risk: $\tilde{\theta} = \text{argmin}_{\hat{\theta} \in \mathbb{R}^d} \sum_{x \in X} (y(x) - \hat{\theta}^T x)^2 + \rho ||\hat{\theta}||_2^2$ , with $\rho > 0$ being the regularization parameter. We claim that simply changing the $\tilde{f}_{-i,t}(\cdot)$ in SR-UCB to the output from the above ridge regression, the $O(\sqrt{\log T / T})$-BNE for inducing an effort level $e^*$ will hold. Different from the non-regularized case, the introduction of the regularization term will add bias in $\tilde{\theta}_{-i}^T(t)$, which gives a biased evaluation of indexes. However, we prove the convergence of $\tilde{\theta}_{-i}^T(t)$ (so again the indexes will converge properly) in the following lemma, which enables an easy adaption of our previous results for non-regularized case to ridge regression:

**Lemma 1.** *With $n$ i.i.d. samples, w.p. $\geq 1 - e^{-Kn}$ ($K > 0$ is a constant), $||\tilde{\theta}_{-i}(t) - \theta||_2^2 \leq O(\frac{1}{n^2})$.*

**Non-linear regression** The basic idea for extending the results to non-linear regression is inspired by the consistency results on $M$-estimator [14], when the error of training data satisfies zero mean. Similar to the reasoning for Lemma 1, if $(\tilde{f}_{-i,t}(x) - f(x))^2 \to 0$, we can hope for an easy adaptation

of our previous results. Suppose the non-linear regression model can be characterized by a parameter family $\Theta$, where $f$ is characterized by parameter $\theta$, and $\tilde{f}_{-i,t}$ by $\tilde{\theta}_i(t)$. Due to the consistency of $M$-estimator we will have $||\tilde{\theta}_i(t) - \theta||_2 \to 0$. More specifically, according to the results from [18], for the non-linear regression model we can establish an $O(1/\sqrt{n})$ convergence rate with $n$ training samples. When $f$ is Lipschitz in parameter space, i.e. there exists a constant $L_N > 0$ such that $|\tilde{f}_{-i,t}(x) - f(x)| \le L_N ||\tilde{\theta}_i(t) - \theta||_2$, by dominated convergence theorem we also have $(\tilde{f}_{-i,t}(x) - f(x))^2 \to 0$, and $(\tilde{f}_{-i,t}(x) - f(x))^2 \le O(1/t)$. The rest of the proof can then follow.

**Example 1.** Logistic function $f(x) = \frac{1}{1+e^{-\theta^T x}}$ satisfies Lipschitz condition with $L_N = 1/4$.

# 6 Computational issues

In order to update the indexes and select workers adaptively, we face a few computational challenges. First, in order to update the index for each worker at any time $t$, a new estimator $\tilde{\theta}_{-i}(t)$ (using data from all other workers $j \ne i$ up to time $t-1$) needs to be re-computed. Second, we need to re-apply $\tilde{\theta}_{-i}(t)$ to every collected sample from worker $i$, $\{(x_i(n), \tilde{y}_i(n, e_i(n)) : i \in d(n), n = 1, 2, ...t-1\}$ from previous rounds. We propose online variants of SR-UCB to address these challenges.

**Online update of $\tilde{\theta}_{-i}(\cdot)$**    Inspired by the online learning literature, instead of re-computing $\tilde{\theta}_{-i}(t)$ at each step, which involves re-calculating the inverse of a covariance matrix (e.g., $(\rho I + X^T X)^{-1}$ for ridge regression) whenever there is a new sample point arriving, we can update $\tilde{\theta}_{-i}(t)$ in an online fashion, which is computationally much more efficient. We demonstrate our results with ridge linear regression. Start with an initial model $\tilde{\theta}_{-i}^{\text{online}}(1)$. Denote by $(x_{-i}(t), \tilde{y}_{-i}(t))$ any newly arrived sample at time $t$ from worker $j \ne i$. Update $\tilde{\theta}_{-i}^{\text{online}}(t+1)$ (for computing $I_i(t+1)$) as [17]:

$$\tilde{\theta}_{-i}^{\text{online}}(t+1) := \tilde{\theta}_{-i}^{\text{online}}(t) - \eta_t \cdot \nabla_{\tilde{\theta}_{-i}^{\text{online}}(t)}[(\theta^T x_{-i}(t) - \tilde{y}_{-i}(t))^2 + \rho||\theta||_2^2],$$

Notice there could be multiple such data points arriving at each time – in which case we will update sequentially in an arbitrarily order. It is also possible that there is no sample point arriving from workers other than $i$ at a time $t$, in which case we simply do not perform an update. Name this online updating SR-UCB as OSR1-UCB. With online updating, the accuracy of trained model $\tilde{\theta}_{-i}^{\text{online}}(t+1)$ converges slower, so is the accuracy in the index for characterizing worker's performance. Nevertheless we prove exerting targeted effort exertion $e^*$ is $O(\sqrt{\log T / T})$-BNE under OSR1-UCB for ridge regression, using convergence results for $\tilde{\theta}_{-i}^{\text{online}}(t)$ proved in [17].

**Online score update**    Online updating can also help compute $S_i(t)$ (in $I_i(t)$) efficiently. Instead of repeatedly re-calculating the score for each data point (in $S_i(t)$), we only update the newly assigned samples which has not been evaluated yet, by replacing $\tilde{\theta}_{-i}^{\text{online}}(t)$ with $\tilde{\theta}_{-i}^{\text{online}}(n)$ in $S_i(t)$:

$$S_i^{\text{online}}(t) := \frac{1}{n_i(t)} \sum_{n=1}^{t} 1(i \in d(n))[a - b((\tilde{\theta}_{-i}^{\text{online}}(n))^T x_i(n) - \tilde{y}_i(n, e_i(n)))^2]. \tag{6.1}$$

With less aggressive update, again the index term's accuracy converges slower than before, which is due to the fact the older data is scored using an older (and less accurate) version of $\tilde{\theta}_{-i}^{\text{online}}$ without being further updated. We propose OSR2-UCB where we change the index SR-UCB to: $S_i^{\text{online}}(t) + c\sqrt{(\log t)^2/n_i(t)}$, to accommondate the slower convergence. We establish an $O(\log T/\sqrt{T})$-BNE for workers exerting target effort—the change is due to the change of the bias term.

# 7 Privacy preserving SR-UCB

With a repeated data acquisition setting, workers' privacy in data may leak repeatedly. In this section we study an extension of SR-UCB to preserve privacy of each individual worker's contributed data. Denote the training data collected as $\mathcal{D} := \{\tilde{y}_i(t, e_i(t))\}_{i \in d(t), t}$. We quantify privacy using differential privacy [5], and we adopt $(\varepsilon, \delta)$-differential privacy (DP) [6], which for our setting is defined below:

**Definition 2.** A mechanism $\mathcal{M} : (X \times \mathbb{R})^{|\mathcal{D}|} \to O$ is $(\varepsilon, \delta)$-differentially private if for any $i \in d(t), t$, any two distinct $\tilde{y}_i(t, e_i(t)), \tilde{y}_i'(t, e_i'(t))$, and for every subset of possible outputs $\mathcal{S} \subseteq O$, $\Pr[\mathcal{M}(\mathcal{D}) \in \mathcal{S}] \le \exp(\varepsilon) \Pr[\mathcal{M}(\mathcal{D}\backslash\{\tilde{y}_i(t, e_i(t))\}, \tilde{y}_i'(t, e_i'(t))) \in \mathcal{S}] + \delta$.

An outcome $o \in O$ of a mechanism contains two parts, both of which can contribute to privacy leakage: (1) The learned regression model $\tilde{\theta}(T)$, which is trained using all data collected after $T$ rounds. Suppose after learning the regression model $\tilde{\theta}(T)$, this information will be released for public usage or monitoring. This information contains each individual worker's private information. Note this is a one-shot leak of privacy (published at the end of the training (step $T$)). (2) The indexes can reveal private information. Each worker $i$'s data will be utilized towards calculating other workers' indexes $I_j(t), j \neq i$, as well as his own $I_i(t)$, which will be published.[7] Note this type of leakage occurs at each step. The lemma below allows us to focus on the privacy losses in $S_j(t)$, instead of $I_j(t)$, as both $I_j(t)$ and $n_i(t)$ are functions of $\{S_j(n)\}_{n \leq t}$.

**Lemma 2.** *At any time $t$, $\forall i$, $n_i(t)$ can be written as a function of $\{S_j(n), n < t\}_j$.*

**Preserving privacy in $\tilde{\theta}(T)$** To protect privacy in $\tilde{\theta}(T)$, following standard method [6], we add a Laplacian noise vector $\mathbf{v}_\theta$ to it: $\tilde{\theta}^p(T) = \tilde{\theta}(T) + \mathbf{v}_\theta$, where $\Pr(\mathbf{v}_\theta) \propto \exp(-\varepsilon_\theta \|\mathbf{v}_\theta\|_2)$. $\varepsilon_\theta > 0$ is a parameter controlling the noise level.

**Lemma 3.** *Set $\varepsilon_\theta = 2\sqrt{T}$, the output $\tilde{\theta}^p(T)$ of SR-UCB for linear regression preserves $(O(T^{-1/2}), \exp(-O(T)))$-DP. Further w.p. $\geq 1 - 1/T^2$, $\|\tilde{\theta}^p(T) - \tilde{\theta}(T)\|_2 = \|\mathbf{v}_\theta\|_2 \leq \log T / \sqrt{T}$.*

**Preserving privacy in $\{I_i(t)\}_{i,t}$: a continual privacy preserving model** For indexes $\{I_i(t)\}_i$, it is also tempting to add $v_i(t)$ to each index, i.e. $I_i(t) := I_i(t) + v_i(t)$, where again $v_i(t)$ is a zero-mean Laplacian noise. However releasing $\{I_i(t)\}_i$ at each step will release a noisy version of each $\tilde{y}_i(n, e_i(n)), i \in d(n), \forall n < t$. The composition theory in differential privacy [12] implies that the preserved privacy level will grow in time $t$, unless we add significant noise on each stage, which will completely destroy the informativeness of our index policy. We borrow the partial sum idea for continual observations [3]. The idea is when releasing continual data, instead of inserting noise at every step, the current to-be-released data will be decoupled into sum of partial sums, and we only add noise to each partial sum and this noisy version of the partial sums can be re-used repeatedly.

We consider adding noise to a modified version of the online indexes $\{S_i^{\text{online}}(t)\}_{i,t}$ as defined in Eqn. (6.1), with $\tilde{\theta}_{-i}^{\text{online}}(t)$ replaced by $\sum_{n=1}^{t} \tilde{\theta}_{-i}(n)/t$, where $\tilde{\theta}_{-i}(n)$ is the regression model we estimated using all data from worker $j \neq i$ up to time $n$. For each worker $i$, his contributed data appear in both $\{S_i^{\text{online}}(t)\}_t$ and $\{S_j^{\text{online}}(t)\}_t, j \neq i$. For $S_j^{\text{online}}(t), j \neq i$, we want to preserve privacy in $\sum_{n=1}^{t} \tilde{\theta}_{-j}(n)/t$, which contains information of $\tilde{y}_i(n, e_i(n))$.

We first apply the partial sums idea to $\sum_{n=1}^{t} \tilde{\theta}_{-j}(n)/t$. Write down $t$ as a binary string and find the rightmost digit that is a 1, then flip that digit to 0: convert is back to decimal gives $q(t)$. Take the sum from $q(t) + 1$ to $t$: $\sum_{n=q(t)+1}^{t} \tilde{\theta}_{-j}(n)$ as one partial sum. Repeat above for $q(t)$, to get $q(q(t))$, and the second partial sum $\sum_{n=q(q(t))+1}^{q(t)} \tilde{\theta}_{-j}(n)$, until we reach $q(\cdot) = 0$. So

$$\sum_{n=1}^{t} \tilde{\theta}_{-j}(n)/t = \frac{1}{t} \left( \sum_{n=q(t)+1}^{t} \tilde{\theta}_{-j}(n) + \sum_{n=q(q(t))+1}^{q(t)} \tilde{\theta}_{-j}(n) + ... + \sum_{n=0}^{0} \tilde{\theta}_{-j}(n) \right). \tag{7.1}$$

Add noise $\mathbf{v}_{\tilde{\theta}}$ with $\Pr(\mathbf{v}_{\tilde{\theta}}) \propto e^{-\varepsilon \|\mathbf{v}_{\tilde{\theta}}\|_2}$ to each partial sum. The number of noise terms is bounded by $\leq \lceil \log t \rceil$ at time $t$. So is the number of appearance of each private data in the partial sums [3]. Denote the noisy version of $\sum_{n=1}^{t} \tilde{\theta}_{-j}(n)/t$ as $\tilde{\tilde{\theta}}_{-i}^{\text{online}}(n)$. Each $S_i^{\text{online}}(t)$ is computed using $\tilde{\tilde{\theta}}_{-i}^{\text{online}}(n)$.

For $S_i^{\text{online}}(t)$, we also want to preserve privacy in $\tilde{y}_i(n, e_i(n))$. Clearly $S_i^{\text{online}}(t)$ can be written as sum of partial sums of terms involving $\tilde{y}_i(n, e_i(n))$: write $S_i^{\text{online}}(t)$ as a summation: $\sum_{n=1}^{n_i(t)} dS(n)/n_i(t)$ (short-handing $dS(n) := a - b((\tilde{\tilde{\theta}}_{-i}^{\text{online}}(t(n)))^T x_i(t(n)) - \tilde{y}_i(t(n), e_i(t(n))))^2$, where $t(n)$ denotes the time of worker $i$ being sampled the $n$-th time.). Decouple $S_i^{\text{online}}(t)$ into partial sums using the same technique. For each partial sum, add a noise $v_S$ with distribution $\Pr(v_S) \propto e^{-\varepsilon|v_S|}$.

We then show that with the above two noise exertion procedures, our index policy SR-UCB will not lose its value in incentivizing effort. In order to prove similar convergence results, we need to modify SR-UCB by changing the index to the following format:

$$I_i(t) = \hat{S}_i^{\text{online}}(t) + c(\log^3 t \log^3 T)/\sqrt{n_i(t)}, \ \tau(t) = O\left((\log^3 t \log^3 T)/\sqrt{t}\right),$$

where $\hat{S}_i^{\text{online}}(t)$ denotes the noisy version of $S_i^{\text{online}}(t)$ with added noises ( $v_S$, $\mathbf{v}_{\tilde{\theta}}$ etc). The change of bias is mainly to incorporate the increased uncertainty level (due to added privacy preserving noise). Denote this mechanism as PSR-UCB, we have:

**Theorem 3.** *Set* $\varepsilon := 1/\log^3 T$ *for added noises (both* $v_S$ *and* $\mathbf{v}_{\tilde{\theta}}$*), PSR-UCB preserves* $(\mathrm{O}(\log^{-1} T), \mathrm{O}(\log^{-1} T))$*-DP for linear regression.*

With homogeneous workers, we similarly can prove exerting effort $\{e_i^*\}_i$ (optimal effort level) is $\mathrm{O}(\log^6 T/\sqrt{T})$-BNE. We can see that, in order to protect privacy in the bandit setting, the approximation term of BNE is worse than before.

**Acknowledgement:** We acknowledge the support of NSF grant CCF-1301976.

## Footnotes

[1]This assumption can be relaxed. See our supplementary materials for the case with homogeneous $\sigma$.

[2]We assume workers have knowledge of how the mechanism sets up this $\gamma$.

[3]Certainly, we can run mechanisms that induce BNE or dominant-strategy equilibrium for one-shot setting, e.g. [2], for every time step. But such solution does not incorporate long-term incentives.

[4]Combing with the results for homogeneous workers, we can again easily extend our results to the case where there are a mixture of homogeneous and heterogenous workers.

[5]It exists when the supports for $\sigma_1(\cdot), \sigma_2(\cdot)$ overlap for a large support range.

[6]Since we only target the top two workers, we can limit the number of acquisitions on each stage to be no more than two, so the number of query does not go beyond $2T$.

[7]It is debatable whether the indexes should be published or not. But revealing decisions on worker selection will also reveal information on the indexes. We consider the more direct scenario – indexes are published.

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
