[Supplementary Material]

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

[8] Or average over $\sigma(e_i(n))$ when different effort levels are chosen at different steps.

[9]This is really a relaxed argument, in fact we need to prove for a $o(1)$-close to $\sigma_1(e_1^*)$.

[10]And we cannot assume we know it as we are learning it.

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

# APPENDIX

## 8 Proofs for Section 5

### 8.1 Boundedness for indexes

*Proof.* We prove the indexes have bounded support:

$$
\begin{aligned}
&(\tilde{\theta}_{-i}^T(t)x_i(n) - \tilde{y}_i(n,e))^2 \\
&\leq (\tilde{\theta}_{-i}^T(t)x_i(n) - y(x_i(n)) - z_i(e))^2 \\
&\leq 2(\tilde{\theta}_{-i}^T(t)x_i(n) - y(x_i(n)))^2 + 2z^2 \\
&\leq 2(\tilde{\theta}_{-i}^T(t)x_i(n) - \theta^T x_i(n))^2 + 2Z^2 \\
&\leq 2\|\tilde{\theta}_{-i}^T(t) - \theta\|_2^2 \|x_i(n)\|_2^2 + 2Z^2 \\
&\leq 8M^2 + 2Z^2
\end{aligned}
$$

$\square$

### 8.2 Intuitions and some results that are needed for proving Theorem 1

In order to analyze our bandit setting, we need to track the evolution of the indexes, which are mainly affected by the change of the "scoring" term $S_i(t)$. In analogy to classical bandit setting, we are hoping to establish a convergence result for $S_i(t)$. Specifically we prove the following results:

**Lemma 4.** *Suppose we have n i.i.d. samples to construct $\tilde{\theta}_{-i}(t)$ in $S_i(t)$. Then*

$$
|\mathbb{E}[S_i(t)] - (a - b(\sigma_z + \sigma(e_i)))| \leq O\left(\frac{1}{n^2}\right).
\tag{8.1}
$$

*And w.p. being at least $1 - e^{-Kn}$ for some $K > 0$,*

$$
|S_i^1(t)| \leq O\left(\frac{1}{n^2}\right).
$$

*Proof.* To give some intuition, we first decouple the quadratic term $(\tilde{\theta}_{-i}^T(t)x_i(n) - \tilde{y}_i(n,e))^2$ in each $S_i(t)$, for any time $t$, and any data sample $x_i(n)$ that is collected before $t$ ($n \leq t$):

$$
\left(\tilde{\theta}_{-i}^T(t)x_i(n) - \tilde{y}_i(n,e)\right)^2 = \left(\tilde{\theta}_{-i}^T(t)x_i(n) - y(x_i(n)) + y(x_i(n)) - \tilde{y}_i(n,e)\right)^2
$$

$$
= (\tilde{\theta}_{-i}^T(t)x_i(n) - y(x_i(n)))^2 + (\tilde{\theta}_{-i}^T(t)x_i(n) - y(x_i(n)))(y(x_i(n)) - \tilde{y}_i(n,e)) + (y(x_i(n)) - \tilde{y}_i(n,e))^2
$$

$$
= \underbrace{((\tilde{\theta}_{-i}(t) - \theta)^T x_i(n))^2}_{l_{i,t}^1(n)} + \underbrace{(\tilde{\theta}_{-i}(t) - \theta)^T x_i(n) \cdot z(n)}_{l_{i,t}^2(n)} + \underbrace{(\tilde{\theta}_{-i}^T(t)x_i(n) - y(x_i(n)))(y(x_i(n)) - \tilde{y}_i(n,e_i))}_{l_{i,t}^3(n)}
$$

$$
+ \underbrace{z^2(n)}_{l_{i,t}^4(n)} + \underbrace{(y(x_i(n)) - \tilde{y}_i(n,e))^2}_{l_{i,t}^5(n)}.
\tag{8.2}
$$

With above decoupling we can re-write $S_i(t)$ as

$$S_i(t) := a + \sum_{k=1}^{5} S_i^k(t), \text{ where } S_i^k(t) = -b \frac{\sum_{n=1}^{t-1} \mathbb{1}(i \in d(n)) l_{i,t}^k(n)}{n_i(t)} \ .$$

We analyze each of the five terms $S_i^k(t), k = 1, 2, ..., 5$. For the first term $l_{i,t}^1(n)$ first notice $\forall n$

$$((\tilde{\theta}_{-i}(t) - \theta)^T x_i(n))^2 \leq ||\tilde{\theta}_{-i}(t) - \theta||_2^2 ||x_i(n)||_2^2 \leq ||\tilde{\theta}_{-i}(t) - \theta||_2^2 \ .$$

We have the following lemma:

**Lemma 5.** *Suppose we have n i.i.d. samples to construct $\tilde{\theta}_{-i}(t)$, then w.p. being at least $1 - e^{-Kn}$ where $K > 0$ is a constant,*

$$||\tilde{\theta}_{-i}(t) - \theta||_2^2 \leq Z^4 (d+2)^6 \frac{(1+\xi)^2}{(1-\xi)^4} \frac{1}{n^2}, \text{ with } \xi \in (0,1) \text{ being a constant.}$$

Using above lemma we know w.p. being at least $1 - e^{-Kn}$,

$$|S_i^1(t)| \leq b||\tilde{\theta}_{-i}(t) - \theta||_2^2 \leq bZ^4 (d+2)^6 \frac{(1+\xi)^2}{(1-\xi)^4} \frac{1}{n^2} \ .$$

For the second term $l_{i,t}^2(n)$, consider its expectation. Due to independence between any data, and the independence between data and noise $z$, we have

$$\mathbb{E}[l_{i,t}^2(n))] = \mathbb{E}(\tilde{\theta}_{-i}(t) - \theta)^T x_i(n)) \mathbb{E}(z) = 0 \ .$$

Similarly for $l_{i,t}^3(n)$, since the noise term in $\tilde{\theta}_{-i}^T(t)$ is independent from the one in $y(x_i(n)) - \tilde{y}_i(n,e)$, again we have

$$\mathbb{E}[l_{i,t}^3(n)] = \mathbb{E}[\tilde{\theta}_{-i}^T(t)x_i(n) - y(x_i(n))] \cdot \mathbb{E}[y(x_i(n)) - \tilde{y}_i(n,e_i)] = 0 \ .$$

The second equality follows as $\mathbb{E}[y(x_i(n)) - \tilde{y}_i(n,e_i)] = 0$. Also we would like to note that due to the boundedness of $(\tilde{\theta}_{-i}(t) - \theta)^T x_i(n)$ and $\tilde{\theta}_{-i}^T(t)x_i(n) - y(x_i(n))$, the convergence of $S_i^2(t), S_i^3(t)$ can be established using Hoeffding bound [5].

For $l_{i,t}^4(n), l_{i,t}^5(n)$ we have (suppose worker $i$ exerts consistent effort $e_i$)

$$\mathbb{E}[l_{i,t}^4(n)] = \mathbb{E}[z^2(n)] = \sigma_z, \ \mathbb{E}[l_{i,t}^5(n)]^2 = \mathbb{E}[z_i(e_i)]^2 = \sigma(e_i) \ .$$

The convergence rate is depending on how many samples worker $i$ has been assigned. To summarize we know

$$\mathbb{E}[\sum_{k=2}^{5} S_j^k(t)] = -b(\sigma_z + \sigma(e_i)). \tag{8.3}$$

And the expected scoring term $S_i$ for each user will roughly converge to

$$a - b(\sigma_z + \sigma(e_i)) + O\left(\frac{1}{n^2}\right), \ , \tag{8.4}$$

with $O\left(\frac{1}{n^2}\right)$ being an additional bias term, where $n$ is the number of samples contributed by other workers. This also implies that

$$|\mathbb{E}[S_i(t)] - (a - b(\sigma_z + \sigma(e_i)))| \leq O\left(\frac{1}{n^2}\right).$$

$\square$

With above preparation we see if every worker is exerting the same level of efforts, $e_i \equiv e$, the expected scoring function for workers will become equivalent. Then in order to be selected, workers will *race* with each other on $\sigma(e_i)$[8] and be incentivized to exert efforts.

## 8.3 Proof for Lemma 5

*Proof.* Denote the stacked data in a matrix form as $\mathbf{X} \in \mathbb{R}^{n \times d}$, and the corresponding labeling outcome $\tilde{y} \in \mathbb{R}^n$. Then it is well known that the optimal estimator from minimizing a non-regularized empirical loss function is given by $\tilde{\theta}_{-i}(t) = (\mathbf{X}^T \mathbf{X})^{-1} \mathbf{X}^T \tilde{y}$. Denote $y$ as the true labels. Consider the following facts.

$$
\begin{aligned}
& ||\tilde{\theta}_{-i}(t) - \theta||_2^2 \\
&= ||(\mathbf{X}^T \mathbf{X})^{-1} \mathbf{X}^T \tilde{y} - (\mathbf{X}^T \mathbf{X})^{-1} \mathbf{X}^T y||_2^2 \\
&= \text{trace}((\mathbf{X}^T \mathbf{X})^{-1} \mathbf{X}^T (\tilde{y} - y)(\tilde{y} - y)^T \mathbf{X} (\mathbf{X}^T \mathbf{X})^{-1}) \\
&= ||(\mathbf{X}^T \mathbf{X})^{-1} \mathbf{X}^T (\tilde{y} - y)(\tilde{y} - y)^T \mathbf{X} (\mathbf{X}^T \mathbf{X})^{-1}||_2^2 \\
&\leq ||(\mathbf{X}^T \mathbf{X})^{-1}||_2^2 \cdot ||\mathbf{X}^T (\tilde{y} - y)(\tilde{y} - y)^T \mathbf{X}||_2^2 \cdot ||(\mathbf{X}^T \mathbf{X})^{-1}||_2^2 .
\end{aligned}
\tag{8.5}
$$

Since $x$s are sampled uniformly from a unit ball, by Theorem 7 in [4] (adapted from Corollary 5.52 in [23]), $||(\mathbf{X}^T \mathbf{X})^{-1}||_2^2$ can be bounded at the order of $O\left(\frac{1}{n^2}\right)$ w.h.p. $(> 1 - O(e^{-Kn}))$:

**Theorem 4.** *Let $\xi \in (0,1)$, and $t \geq 1$. Let $|| \cdot ||$ denote the spectral norm. If $\{x_i\}_{i=1}^n$ are i.i.d. and sample uniformly from the unit ball (with dimension $d$), then w.p. being at least $1 - d^{-t^2}$, when $n \geq C(\frac{t}{\xi})^2(d+2) \log d$, for some constant $C$, then*

$$
||\mathbf{X}^T \mathbf{X}|| \leq \frac{1+\xi}{2+d}n, \ ||(\mathbf{X}^T \mathbf{X})^{-1}|| \leq \frac{1}{(1-\xi)\frac{1}{2+d}n} .
\tag{8.6}
$$

We will be repeatedly using this lemma. Then the first and third term in Eqn. (8.5) can be well bounded: $||(\mathbf{X}^T \mathbf{X})^{-1}||_2^2 \leq \frac{1}{(1-\xi)\frac{1}{2+d}n}$ . Consider the second term. For $||\mathbf{X}^T (\tilde{y} - y)(\tilde{y} - y)^T \mathbf{X}||_2^2$, w.h.p.,

$$
\begin{aligned}
||\mathbf{X}^T (\tilde{y} - y)(\tilde{y} - y)^T \mathbf{X}||_2^2 &= \text{trace}(\mathbf{X}^T (\tilde{y} - y)(\tilde{y} - y)^T \mathbf{X}) \\
&\leq \max_i z_i^2 \cdot \sum_i x_i^T x_i = \max_i z_i^2 \cdot \text{trace}(\mathbf{X}^T \mathbf{X}) \\
&\leq Z^2 \cdot \text{trace}(\mathbf{X}^T \mathbf{X}) = Z^2 |||\mathbf{X}^T \mathbf{X}||_2^2 \\
&\leq Z^2 \frac{1+\xi}{2+d}n ,
\end{aligned}
$$

Combining above argument, we establish that w.h.p.,

$$
\begin{aligned}
||\tilde{\theta}_{-i}(t) - \theta||_2^2 &\leq \left(\frac{1}{(1-\xi)\frac{1}{2+d}n}\right)^2 \cdot \left(Z^2 \frac{1+\xi}{2+d}n\right)^2 \cdot \left(\frac{1}{(1-\xi)\frac{1}{2+d}n}\right)^2 \\
&= Z^4 (d+2)^6 \frac{(1+\xi)^2}{(1-\xi)^4} \frac{1}{n^2} \to 0, \text{ as } n \to \infty.
\end{aligned}
$$

$\square$

## 8.4 Proof for Theorem 1

*Proof.* To prove the theorem, we proceed in the following ways. We first prove the following lemma:

**Lemma 6.** *If every worker exerts effort level $e_i(t) = e^*, \forall t$, there exists a constant $\delta_U > 0$ such that for any $i, j$ that $i \neq j$ we have probability at least $1 - O\left(\frac{1}{t^2}\right)$, $n_i(t) \leq (1 + \delta_U)n_j(t)$.*

What this lemma is implying is that w.h.p., one worker cannot be selected more than another by a constant fraction. This result is crucial for us to establish the index analysis for bandits – different from classical bandit, due to the lack of ground-truth, the evaluation of each worker's index does

not only depend on the number of samples from worker himself, but also on the ones from other workers. With above results at hand, and using union bound we know w.p. being at least $1 - O\left(\frac{N}{t^2}\right)$,

$$(\frac{N-1}{1+\delta_U}+1)n_i(t) \leq \sum_j n_j(t) \leq [(1+\delta_U)(N-1)+1]n_i(t).$$

Since $\sum_j n_j(t) \geq t$ (at least one selection at each time) we must have $n_i(t) \geq \frac{t}{(1+\delta_U)(N-1)+1}$. Based on this we can now establish the following lemma.

**Lemma 7.** *If every worker exerts effort level $e_i(t) = e^*, \forall t$, we have $\mathbb{E}[n_i(t)] \geq t - const.$ .*

With this lemma we are most ready to prove the first part of the $\pi$−BNE. First of all, for any worker, there is no reason to deviate to $e > e^*$. This is due to the fact with exerting $e^*$ each worker has already guaranteed nearly $T$ number of selection. Further exerting effort, while will decrease the net payment at each step, will at most bring in $O\left(\frac{1}{T}\right)$ gain per round (a constant number more selections).

Now we show deviating to $e < e^* - O\left(\sqrt{\frac{\log T}{T^z}}\right), \forall 0 \leq z < 1$ will also be non-profitable. For any such $z$, we can always find a $z' = z + \varsigma < 1$, $\varsigma > 0$ such that $\sqrt{\frac{\log t}{t^{z'}}} < \sqrt{\frac{\log t}{t^z}}$. Denote $\Delta := O\left(\sqrt{\frac{\log T}{T^z}}\right)$. Therefore we will be having (using convexity and smoothness of $\sigma$)

$$\sigma(e - \Delta) \geq \sigma(e) - \sigma'(e)(-\Delta) \geq \sigma(e) + L\Delta$$
$$\Rightarrow \sigma(e - \Delta) - \sigma(e) \geq \underline{L}\Delta. \tag{8.7}$$

This creates $b\underline{L}\Delta$ difference in $\mathbb{E}[\sum_{k=2}^5 S_j^k(t)]$ based on Eqn.(8.3) (exerting $e^*$ and $e$). Suppose worker $i$ is deviating, then we prove:

**Lemma 8.** *After $O\left(T^{z'}\right)$ selections, the number of selection of worker $i$ can be bounded as follows*

$$\mathbb{E}[n_i(T; t \geq T^{z'})] \leq O\left(\frac{\log T}{\Delta^2}\right) = O\left(T^z\right).$$

Then by deviating the number of selection of worker $i$ is bounded by $\max\{O\left(T^{z'}\right), O\left(T^z\right)\} \leq O\left(T^{z'}\right)$. Following which we know the collected reward for worker $i$ is then upper bounded by $(\gamma + \Delta) \cdot O\left(T^{z'}\right) < \gamma \cdot O(T)$, when $\gamma = \Omega\left(\sqrt{\frac{\log T}{T}}\right)$, and $T$ is large, and $z'$ is selected such that $z < z' < \frac{z+1}{2}$ . On the other hand, when the deviation is no more than $O\left(\sqrt{\frac{\log T}{T}}\right)$, the per round gain is bounded by

$$\underbrace{O\left(\sqrt{\frac{\log T}{T}}\right) + \gamma}_{\text{after deviation}} - \underbrace{\frac{\gamma \cdot (T - const.)}{T}}_{\text{before deviation}} \leq O\left(\sqrt{\frac{\log T}{T}}\right),$$

Thus the above argument establishes that a consistent deviation will result in at most $\sqrt{\frac{\log T}{T}}$ more net-payment per task.

We now prove the case when workers may deviate differently at different step. Take worker $i$ as an example, denote its effort level at step $t$ as $e_i(t)$. Denote $\Delta_i(t) = e^* - e_i(t) \geq 0$ as a per-step deviation. First we have by convexity $\sigma(e_i(t)) - \sigma(e^*) \geq \sigma'(e^*)\Delta_i(t)$ . Sum over all period of time we have

$$\frac{\sum_{t=1}^T \sigma(e_i(t))}{T} - \sigma(e^*) \geq \sigma'(e^*)\frac{\sum_{t=1}^T \Delta_i(t)}{T} . \tag{8.8}$$

If $\frac{\sum_{t=1}^T \Delta_i(t)}{T} \leq 0$, we know that the total cost is higher than $Te^*$. Then

$$\frac{\sum_{t=1}^T \mathbb{1}(i \in d(t))(p_i - e_i(t)) - E[n_i(t, e^*)]\gamma}{T}$$
$$\leq \frac{\sum_{t=1}^T \mathbb{1}(i \in d(t))(e_i + \gamma - e_i(t)) - (T - const.)\gamma}{T}$$
$$\leq \frac{const.}{T}\gamma + \frac{\sum_{t=1}^T \mathbb{1}(i \in d(t))\Delta_i(t)}{T} \leq \frac{const.}{T}\gamma.$$

So the per-round profit is upper bounded by $\frac{\text{const.}}{T}\gamma$ by such a deviation. Now consider the case $\frac{\sum_{t=1}^{T}\Delta_i(t)}{T}>0$. We then have

$$\frac{\sum_{t=1}^{T}\sigma(e_i(t))}{T}-\sigma(e^*)\geq \underline{L}\frac{\sum_{t=1}^{T}\Delta_i(t)}{T}\ . \tag{8.9}$$

Denote by $\Delta := \underline{L}\frac{\sum_{t=1}^{T}\Delta_i(t)}{T}>0$. Denote by $t'-1$ the last time such that

$$\underline{L}\frac{\sum_{t=1}^{t'-1}\Delta_i(t)}{t'-1}<\Delta/2.$$

If there does not exist such a $t'$, that is for all $t'$ $\underline{L}\frac{\sum_{t=1}^{t'-1}\Delta_i(t)}{t'-1}\geq \Delta/2$, we simply set $t'=1$. Then starting from $t'$, we have

$$\underline{L}\frac{\sum_{t=1}^{t'}\Delta_i(t)}{t'}\geq \Delta/2.$$

When $\Delta = \mathrm{O}\big(\sqrt{\log T/T^z}\big)$, $z\to 1$, we discuss in three cases.

- **Case 1:** When $t'\geq T-\mathrm{O}(\sqrt{T})$. We must have $\underline{L}\frac{\sum_{t=1}^{t'}\Delta_i(t)}{t'}\leq 2\Delta/3$, as otherwise

$$\underline{L}\frac{\sum_{t=1}^{t'-1}\Delta_i(t)}{t'-1}\geq \underline{L}\frac{\sum_{t=1}^{t'-1}\Delta_i(t)}{t'}\geq \underline{L}\frac{\sum_{t=1}^{t'}\Delta_i(t)-\bar{e}}{t'}\geq 2\Delta/3-\frac{\bar{e}}{t'}\geq \Delta/2,$$

which contradicts the definition of $t'$. Then for this case, the average utility gain is upper bounded by the following case (being selected for all the rest of $\mathrm{O}(\sqrt{T})$ steps): $2\Delta/3+\mathrm{O}(\sqrt{T}/T)$. So this establishes the $\mathrm{O}\big(\sqrt{\log T/T}\big)$-BNE.

- **Case 2:** For the second case that $t'=o(T)$, specifically say $t'=\mathrm{O}(T^z)$, $0<z<1$. We can prove a result that is similar to Lemma 8 stating that

$$\mathbb{E}[n_i(T;t\geq T^{z'})]\leq \mathrm{O}\big(\frac{\log T}{\Delta^2}\big)\ .$$

All previous analysis establishes themselves directly except for the convergence of the fifth term $S_i^5(t)$, as now it consists of non-identical noise terms. Nevertheless using Hoeffding bound, we can establish the convergence of the sum of sequence of non-identical but independent samples $S_i^5(t)\to \frac{\sum_{t=1}^{T}\sigma(e_i(t))}{T}$. If $\frac{\sum_{t=1}^{T}\Delta_i(t)}{T}=\Omega(\sqrt{\frac{\log T}{T^z}})$, we will again have

$$\frac{\sum_{t=1}^{T}\sigma(e_i(t))}{T}-\sigma(e^*)=\Omega(\sqrt{\frac{\log T}{T^z}})\ , \tag{8.10}$$

from which we can prove a contradiction on profitable deviations, via similarly proving the bound on the number of selection (Lemma (8)), i.e., by deviating the number of selection of worker $i$ is bounded by $\max\{\mathrm{O}(T^{z'}),\mathrm{O}(T^z)\}=\mathrm{O}(T^{z'})$; and the rest analysis follows.

- **Case 3:** For the third case that $\mathrm{O}(T^z)\leq t'\leq T-\mathrm{O}(\sqrt{T})$. Again we must have $\underline{L}\frac{\sum_{t=1}^{t'}\Delta_i(t)}{t'}\leq 2\Delta/3$, as otherwise

$$\underline{L}\frac{\sum_{t=1}^{t'-1}\Delta_i(t)}{t'-1}\geq \underline{L}\frac{\sum_{t=1}^{t'-1}\Delta_i(t)}{t'}\geq \underline{L}\frac{\sum_{t=1}^{t'}\Delta_i(t)-\bar{e}}{t'}\geq 2\Delta/3-\mathrm{O}(1/T^z)\geq \Delta/2,$$

as $\mathrm{O}(1/T^z)\leq \sqrt{\log T/T^z}$. Then we can repeat the argument for **Case 2**, but with a deviation analysis on the interval of $[\mathrm{O}(T^z),T]$, with the starting time being $\mathrm{O}(T^z)$ or larger. Then similar to the case with $t'=o(T)$ (as now $\mathrm{O}(T^z)$ is as if $t'=1$), we can prove that $\mathbb{E}[n_i(T;t\geq T^{z'})]\leq \mathrm{O}\big(\frac{\log T}{\Delta^2}\big)$. Yet the average gain per step before $t'$ is bounded by $2\Delta/3=\mathrm{O}\big(\sqrt{\frac{\log T}{T^z}}\big)$.

When $\Delta > \sqrt{\frac{\log T}{T}}$, we will take $t'$ as the last time that $\underline{L}\frac{\sum_{t=1}^{t'-1}\Delta_i(t)}{t'-1}<\Delta/4$ instead. This argument repeats by above halfing procedure until $\Delta$ reduces to the order of $\sqrt{\frac{\log T}{T}}$, and $t'$ will remain $o(T)$. Then the above argument can be applied. Combine all above we proved the theorem. □

## 8.5 Proof of Lemma 6

*Proof.* We follow the notations and definitions in Section 8.2 and Lemma 4 therein ($S_i^k$ etc). Suppose at a certain time $t$ we have $n_i(t) = (1 + \delta_U) n_j(t), \delta_U > 0$. We would like to bound the following probability $\Pr[I_i(t) \geq I_j(t)]$. This is equivalent with proving the following:

$$\Pr[I_i(t) \geq I_j(t)] = \Pr\left[S_i(t) + c\sqrt{\frac{\log t}{n_i(t)}} \geq S_j(t) + c\sqrt{\frac{\log t}{n_i(t)}}\right]$$

$$= \Pr\left[S_i(t) + c\sqrt{\frac{\log t}{n_i(t)}} \geq S_j(t) + c\sqrt{\frac{(1+\delta_U)\log t}{n_j(t)}}\right]$$

$$= \Pr\left[S_i(t) - S_j(t) \geq (\sqrt{1+\delta_U} - 1)c\sqrt{\frac{\log t}{n_i(t)}}\right].$$

Using Lemma 5, and denote $C_1 := bZ^4(d+2)^6 \frac{(1+\xi)^2}{(1-\xi)^4}$. Then we know with probability at least $1 - e^{-K\sum_{k \neq i} n_k(t)}$, and $1 - e^{-K\sum_{k \neq j} n_k(t)}$ (with $K > 0$ being a constant) respectively (when worker $i, j$ exert effort levels $e_i, e_j$ respectively),

$$|S_i^1(t)| \leq \frac{C_1}{(\sum_{k \neq i} n_k(t))^2}, \quad |S_j^1(t)| \leq \frac{C_1}{(\sum_{k \neq j} n_k(t))^2},$$

For $\sum_{k \neq i} n_k(t)$ we discuss two cases. For the first case, if there exists a constant $\nu$ such that $n_i(t) \leq (1-\nu)t$, then $\sum_{k \neq i} n_k(t) \geq \nu t$. Otherwise if $n_i(t) > (1-\nu)t$ we will also have

$$\sum_{k \neq i} n_k(t) \geq n_j(t) \geq \frac{n_i(t)}{1 + \delta_U} \geq \frac{1 - \nu}{1 + \delta_U} t$$

so to summarize

$$\sum_{k \neq i} n_k(t) \geq \min\{\nu, \frac{1 - \nu}{1 + \delta_U}\} t.$$

Similarly we can prove that

$$\sum_{k \neq j} n_k(t) \geq \min\{\nu, (1-\nu)(1 + \delta_U)\} t.$$

Denote as $C_2 = \min\{\nu, \frac{1-\nu}{1+\delta_U}, (1-\nu)(1 + \delta_U)\}$. We will have with probability at least $1 - e^{-KC_2 t}$

$$\max\{|S_i^1(t)|, |S_j^1(t)|\} \leq \frac{C_1}{C_2^2 t^2}.$$

Then

$$\Pr\left[S_i(t) - S_j(t) \geq (\sqrt{1+\delta_U} - 1)c\sqrt{\frac{\log t}{n_i(t)}}\right]$$

$$\leq \Pr\left[\sum_{k=2}^{5} S_i^k(t) - \sum_{k=2}^{5} S_j^k(t) \geq (\sqrt{1+\delta_U} - 1)c\sqrt{\frac{\log t}{n_i(t)}} - \frac{2C_1}{C_2^2 t^2}\right].$$

Since $\mathbb{E}[\sum_{k=2}^{5} S_i^k(t)] = \mathbb{E}[\sum_{k=2}^{5} S_j^k(t)]$ (at equilibria, and worker $i$ is also exerting the same amount of effort), using union bound, the above implies that

$$\Pr\left[\sum_{k=2}^{5} S_i^k(t) - \sum_{k=2}^{5} S_i^j(t) \geq (\sqrt{1+\delta_U} - 1)c\sqrt{\frac{\log t}{n_i(t)}} - \frac{2C_1}{C_2^2 t^2}\right]$$

$$\leq \Pr\left[\sum_{k=2}^{5} S_i^k(t) - \mathbb{E}[\sum_{k=2}^{5} S_i^k(t)] \geq \frac{\sqrt{1+\delta_U} - 1}{2}c\sqrt{\frac{\log t}{n_i(t)}} - \frac{C_1}{C_2^2 t^2}\right]$$

$$+ \Pr\left[\sum_{k=2}^{5} S_j^k(t) - \mathbb{E}[\sum_{k=2}^{5} S_j^k(t)] \leq \frac{\sqrt{1+\delta_U} - 1}{2}c\sqrt{\frac{\log t}{n_i(t)}} - \frac{C_1}{C_2^2 t^2}\right]. \quad (8.11)$$

We bound each of above two terms. (Due to symmetry we only show the bound for one of them.) For worker $i$, via union bound:

$$\Pr\left[\sum_{k=2}^{5} S_i^k(t) - \mathbb{E}[\sum_{k=2}^{5} S_i^k(t)] \geq \frac{\sqrt{1+\delta_U}-1}{2}c\sqrt{\frac{\log t}{n_i(t)}} - \frac{C_1}{C_2^2 t^2}\right]$$

$$\leq \sum_{k=2}^{5} \Pr\left[S_i^k(t) - \mathbb{E}[S_i^k(t)] \geq \frac{\sqrt{1+\delta_U}-1}{8}c\sqrt{\frac{\log t}{n_i(t)}} - \frac{C_1}{4C_2^2 t^2}\right].$$

Since $n_i(t) \leq t$ we know when $t$ is large $\sqrt{\frac{\log t}{n_i(t)}} \geq \sqrt{\frac{\log t}{t}} \geq \frac{1}{t^2}$. So when $c$ is large enough, e.g. $\frac{\sqrt{1+\delta_U}-1}{8}c > \frac{C_1}{4C_2}$, we will be having

$$\frac{\sqrt{1+\delta_U}-1}{8}c\sqrt{\frac{\log t}{n_i(t)}} - \frac{C_1}{4C_2^2 t^2} > 0.$$

$S_i^2(t), S_i^3(t)$ can be bounded similarly, while $S_i^4(t), S_i^5(t)$ share similar concentration bound. W.l.o.g., we show the derivation for one of each pair. For $S_i^2(t)$, first of all notice

$$|b \cdot l_{i,t}^2(n)| = |b(\tilde{\theta}_{-i}(t) - \theta)^T x_i(n) \cdot z(n)| \leq b||\tilde{\theta}_{-i}(t) - \theta||_2 ||x_i(n)||_2 |z(n)|$$

$$\leq b(||\tilde{\theta}_{-i}(t)||_2 + ||\theta||_2)Z \leq 2MZ$$

$$\Rightarrow -2bMZ \leq l_{i,t}^2(n) \leq 2bMZ.$$

Then via Hoeffding inequality we know

$$\Pr\left[S_i^2(t) - E[S_i^2(t)] \geq \frac{\sqrt{1+\delta_U}-1}{8}c\sqrt{\frac{\log t}{n_i(t)}} - \frac{C_1}{4C_2^2 t^2}\right]$$

$$\leq \exp\left(-\frac{2(\frac{\sqrt{1+\delta_U}-1}{8}c\sqrt{\frac{\log t}{n_i(t)}} - \frac{C_1}{4C_2^2 t^2})^2 n_i^2(t)}{16b^2 M^2 Z^2 n_i(t)}\right)$$

$$\leq \exp\left(-\frac{2(\frac{\sqrt{1+\delta_U}-1}{8}c\sqrt{\frac{\log t}{n_i(t)}} - \frac{C_1}{4C_2^2 t^2})^2 n_i(t)}{16b^2 M^2 Z^2}\right)$$

$$\leq \exp\left(-2\frac{(\frac{\sqrt{1+\delta_U}-1}{8})^2 c^2}{16b^2 M^2 Z^2}\log t\right) \cdot \exp\left(2\frac{\frac{\sqrt{1+\delta_U}-1}{8}c \cdot \frac{C_1}{4C_2^2 t^2}\sqrt{\log t \cdot n_i(t)}}{16b^2 M^2 Z^2}\right)$$

$$\leq \exp\left(-2\frac{(\frac{\sqrt{1+\delta_U}-1}{8})^2 c^2}{16b^2 M^2 Z^2}\log t\right) \cdot \exp\left(2\frac{\frac{\sqrt{1+\delta_U}-1}{8}c \cdot \frac{C_1}{4C_2^2 t}}{16b^2 M^2 Z^2}\right)$$

$$\leq \frac{1}{t^2} \cdot \exp(2/t) \leq \frac{2}{t^2},$$

when $\delta_U$ and $c$ are selected to be large enough, and $t$ large enough: for example

$$\frac{\sqrt{1+\delta_U}-1}{8}c \geq 4bMZ \cdot \max\{1, \frac{4C_2^2}{C_1}\}, \text{ and } t \geq 4.$$

Similarly we can bound $S_i^3(t)$. Now consider $S_i^4(t)$. We use Hoeffding bound via first observing the boundedness of each term $bl_{i,t}^4(n) = |bz^2(n)| \Rightarrow 0 \le b \cdot l_{i,t}^4(n) \le bZ^2$. Then

$$\Pr\left[S_i^4(t) - \mathbb{E}[S_i^4(t)] \ge \frac{\sqrt{1+\delta_U}-1}{8}c\sqrt{\frac{\log t}{n_i(t)}} - \frac{C_1}{4C_2^2 t^2}\right]$$

$$\le \exp\left(-\frac{2((\frac{\sqrt{1+\delta_U}-1}{8})c\sqrt{\frac{\log t}{n_i(t)}} - \frac{C}{4C_2^2 t^2})n_i^2(t)}{b^2 Z^4 n_i(t)}\right)$$

$$\le \exp\left(-2\frac{(\frac{\sqrt{1+\delta_U}-1}{8})^2 c^2}{b^2 Z^4}\log t\right) \cdot \exp\left(2\frac{\frac{\sqrt{1+\delta_U}-1}{8}c \cdot \frac{C_1}{4C_2^2 t^2}\sqrt{\log t \cdot n_i(t)}}{b^2 Z^4}\right)$$

$$\le \exp\left(-2\frac{(\frac{\sqrt{1+\delta_U}-1}{8})^2 c^2}{b^2 Z^4}\log t\right) \cdot \exp\left(2\frac{\frac{\sqrt{1+\delta_U}-1}{8}c \cdot \frac{C_1}{4C_2^2 t}}{b^2 Z^4}\right)$$

$$\le \frac{1}{t^2} \cdot e^{2/t} \le \frac{2}{t^2},$$

again when $\delta_U$ and $c$ are selected to be large enough, and $t$ large enough: for example

$$\frac{\sqrt{1+\delta_U}-1}{8}c \ge bZ^2 \cdot \max\{1, \frac{4C_2^2}{C_1}\}, \text{ and } t \ge 4.$$

Similarly we can bound the term invoking $S_i^4(t)$. Also similarly we can bound

$$\Pr\left[S_j(t) - \mathbb{E}[S_j(t)] \le (\frac{\sqrt{1+\delta_U}-1}{2})c\sqrt{\frac{\log t}{n_i(t)}} - \frac{C}{C_2^2 t^2}\right] \le O\left(\frac{1}{t^2}\right).$$

And in all summarize we proved $\Pr[I_i(t) \ge I_j(t)] \le O\left(\frac{1}{t^2}\right)$.

Now at time $t$, if $n_i(t) > (1+\delta_U)n_j(t)$, we must have a time point $t'$ that $n_i(t')$ changes from $\le (1+\delta_U)n_j(t')$ to $> (1+\delta_U)n_j(t')$, where we must have $n_i(t') \ge (1+\delta_U)n_j(t') - 1 \ge (1+\delta_U - 1)n_j(t')$. Choose $\delta_U$ large enough so $\delta_U - 1$ also satisfies the above claim that $\Pr[I_i(t) \ge I_j(t)] \le O\left(\frac{1}{t^2}\right)$.. We know at time $t'$, it must be $i$ is selected but not $j$, otherwise the ratio between them can only go down (both being selected will not increase a $> 1$ ratio), i.e., it must be $I_i(t') \ge I_j(t')$. We discuss two cases. When $t' \in [t/2, t]$, we know this is upper bounded by $O\left(\frac{1}{(t/2)^2}\right) = O\left(\frac{1}{t^2}\right)$.

If not, consider the worker who has been selected most of the times between $[t/2, t]$. Denote it as $k$. Then we must have $n_k(t) \ge \frac{t}{N2}$. If $n_k(t) \le (1+\delta_U)n_j(t)$, we will have $n_i(t) \ge \frac{t}{2\delta_U N}$, so $n_i(t)/n_j(t) \le \frac{t/2}{\frac{t}{2\delta_U N}} = \delta_U \cdot N$. Otherwise if $n_k(t) > (1+\delta_U)n_j(t)$. We must have there exists a $t'$ such that $t' \ge t/2$ and

$$n_k(t') \ge (1+\delta_U)n_j(t') - 1 \ge (1+\delta_U - 1)n_j(t'),$$

and such that $k$ is selected but not $j$. However we know the probability for this event is also upper bounded by $O\left(1/t^2\right)$. Reset $\delta_U := \delta_U \cdot N$ we finished the proof. $\qquad\square$

## 8.6 Proof for Lemma 7

*Proof.* Following Lemma 6 we know w.h.p. $(\ge 1 - O\left(\frac{1}{t^2}\right))$

$$n_i(t) \ge \frac{t}{(1+\delta_U)(N-1)+1}.$$

Following proof for Lemma 6 we know that w.h.p. $(\ge 1 - O\left(\frac{1}{t^2}\right))$,

$$|S_i(t) - \mathbb{E}[S_i(t)]| \le \frac{\sqrt{1+\delta_U}-1}{2}c\sqrt{\frac{\log t}{n_i(t)}} + \frac{2C_1}{C_2^2 t^2}.$$

Plug in $n_i(t) \geq \frac{t}{(1+\delta_U)(N-1)+1}$ we have

$$|I_i(t) - (a - b(\sigma_z + \sigma(e^*)))| = |S_i(t) + c\sqrt{\frac{\log t}{n_i(t)}} - (a - b(\sigma_z + \sigma(e^*)))|$$

$$\leq |S_i(t) - \mathbb{E}[S_i(t)]| + |\mathbb{E}[S_i(t)] - (a - b(\sigma_z + \sigma(e^*)))| + c\sqrt{\frac{\log t}{n_i(t)}}$$

$$\leq \frac{\sqrt{1+\delta_U}+1}{2} c\sqrt{(1+\delta_U)(N-1)+1}\sqrt{\frac{\log t}{t}} + \frac{3C_1}{C_2^2 t^2}$$

$$\leq \frac{\sqrt{1+\delta_U}+1}{2} c\sqrt{(1+\delta_U)N}\sqrt{\frac{\log t}{t}} + \frac{3C_1}{C_2^2 t^2} \ .$$

Then if we set $\tau(t)$ to be two times of above bound:

$$\tau(t) := 2\left(\frac{\sqrt{1+\delta_U}+1}{2} c\sqrt{(1+\delta_U)N}\sqrt{\frac{\log t}{t}} + \frac{3C_1}{C_2^2 t^2}\right)$$

$$= (\sqrt{1+\delta_U}+1)c\sqrt{(1+\delta_U)N}\sqrt{\frac{\log t}{t}} + \frac{6C_1}{C_2^2 t^2}.$$

we will have

$$\Pr\left[I_j(t) \geq \max_i I_i(t) - \tau(t)\right] \leq O\left(\frac{1}{t^2}\right), \text{ i.e., } \Pr[j \in d(t)] \geq 1 - O\left(\frac{1}{t^2}\right), \forall j, t \ .$$

Therefore we know

$$\mathbb{E}[n_i(T)] = \mathbb{E}[\sum_{n=1}^{T} \mathbb{1}(i \in d(n))] = \sum_{n=1}^{T} \Pr[i \in d(n)] \geq T - O\left(\sum_{n=1}^{T} \frac{1}{n^2}\right) \geq T - \text{const.} \ .$$

$\square$

## 8.7 Proof for Lemma 8

*Proof.* To bound the number of selections of worker $i$ we need to bound $\Pr[I_i(t) \geq \max_j I_j(t) - \tau(t)]$. We further bound this term by the following term $\forall j \neq i, j \in \{1, 2\}$ (top 2 workers):

$$\Pr[I_i(t) \geq I_j(t) - \tau(t)] = \Pr\left[S_i(t) + c\sqrt{\frac{\log t}{n_i(t)}} \geq S_j(t) + c\sqrt{\frac{\log t}{n_j(t)}} - \tau(t)\right].$$

Notice the event $\{S_i(t) + c\sqrt{\frac{\log t}{n_i(t)}} \geq S_j(t) + c\sqrt{\frac{\log t}{n_j(t)}} - \tau(t)\}$ implies at least one of the following should hold

$$\sum_{k=2}^{5} S_i^k(t) - \mathbb{E}[\sum_{k=2}^{5} S_i^k(t)] \geq c\sqrt{\frac{\log t}{n_i(t)}}, \quad \sum_{k=2}^{5} S_j^k(t) - \mathbb{E}[\sum_{k=2}^{5} S_j^k(t)] \leq -c\sqrt{\frac{\log t}{n_i(t)}}$$

$$b\underline{L}\Delta \leq 2c\sqrt{\frac{\log t}{n_i(t)}} + \tau(t) + \frac{C_1}{(\sum_{k\neq i} n_k(t))^2} + \frac{C_1}{(\sum_{k\neq j} n_k(t))^2} \ .$$

As otherwise we will have

$$S_j(t) + c\sqrt{\frac{\log t}{n_j(t)}} - \tau(t) > \sum_{k=2}^{5} S_j^k(t) + c\sqrt{\frac{\log t}{n_j(t)}} - \tau(t) - \frac{C_1}{(\sum_{k\neq j} n_k(t))^2}$$

$$> \mathbb{E}[\sum_{k=2}^{5} S_j^k(t)] - \tau(t) - \frac{C_1}{(\sum_{k\neq j} n_k(t))^2} \geq \mathbb{E}[\sum_{k=2}^{5} S_i^k(t)] + b\bar{L}\Delta - \tau(t) - \frac{C_1}{(\sum_{k\neq j} n_k(t))^2}$$

$$> \mathbb{E}[\sum_{k=2}^{5} S_i^k(t)] + 2c\sqrt{\frac{\log t}{n_i(t)}} + \frac{C_1}{(\sum_{k\neq i} n_k(t))^2} > \sum_{k=2}^{5} S_i^k(t) - c\sqrt{\frac{\log t}{n_i(t)}} + 2c\sqrt{\frac{\log t}{n_i(t)}} + \frac{C_1}{(\sum_{k\neq i} n_k(t))^2}$$

$$= \sum_{k=2}^{5} S_i^k(t) + c\sqrt{\frac{\log t}{n_i(t)}} + \frac{C_1}{(\sum_{k\neq i} n_k(t))^2} \geq S_i(t) + c\sqrt{\frac{\log t}{n_i(t)}} \ ,$$

which is a contradiction. Similarly as in the proof for Lemma 6 we can prove

$$\Pr\left[|\sum_{k=2}^{5} S_i^k(t) - \mathbb{E}[\sum_{k=2}^{5} S_i^k(t)]| \geq c\sqrt{\frac{\log t}{n_i(t)}}\right] = \mathrm{O}(\frac{1}{t^4}),$$

Then when $n_i(t) \geq \mathrm{O}(T^{z'})$, we will be having

$$\sum_{k \neq j} n_k(t) \geq n_i(t) \geq \mathrm{O}(T^{z'}),$$

also repeating argument in the proof for Lemma 6 to establish that $n_i(t) \leq (1 + \delta_U) n_j(t), j \neq i$ (intuitively, $S_i$ converges to a smaller quantity, due to lack of effort. So this side of inequality holds; particularly Eqn.(8.11) holds. We omit the details). So

$$\sum_{k \neq i} n_k(t) \geq n_j(t) \geq \mathrm{O}(T^{z'}),$$

Thus w.p. at least $1 - e^{K\mathrm{O}(T^{z'})} \geq 1 - \mathrm{O}(\frac{1}{t^4})$ (as $T^{z'} \geq \log T \geq \log t$ when $T, z'$ are large) ,

$$2c\sqrt{\frac{\log t}{n_i(t)}} + \tau(t) + \frac{C_1}{(\sum_{k \neq i} n_k(t))^2} + \frac{C_1}{(\sum_{k \neq j} n_k(t))^2} \leq 2c\sqrt{\frac{\log t}{n_i(t)}} + \tau(t) + \mathrm{O}(\frac{1}{T^{2z'}}) \ .$$

Note this is a much smaller quantity compared with $\mathrm{O}(\sqrt{\frac{\log T}{T^z}})$ (since $z' > z$, and this is the amount of deviation). When $t, T$ are larger than certain constants such that $\tau(t) + \mathrm{O}(\frac{1}{T^{2z'}}) < \frac{b\underline{L}\Delta}{2}$, and when $n_i(t) \geq \frac{(2c)^2 \log t}{(\frac{b\underline{L}\Delta}{2})^2}$:

$$2c\sqrt{\frac{\log t}{n_i(t)}} + \tau(t) + \frac{C_1}{(\sum_{k \neq i} n_k(t))^2} + \frac{C_1}{(\sum_{k \neq j} n_k(t))^2} < b\underline{L}\Delta \ .$$

Combined above we know

$$\Pr[I_i(t) \geq \max_j I_j(t) - \tau(t), n_i(t) \geq \frac{(2c)^2 \log t}{(\frac{b\underline{L}\Delta}{2})^2}] = \mathrm{O}(\frac{1}{t^4}) \ .$$

That is after $n_i(t) \geq \frac{(2c)^2 \log t}{(\frac{b\underline{L}\Delta}{2})^2}$ number of selections, worker $i$ will not be selected, except for the $\mathrm{O}(\frac{1}{t^4})$ fraction of probability. Then following the classical method detailed in [1] for UCB1 (the three way arguments), we know the expected number of selection $\mathbb{E}[n_i(T)]$ bounds as follows: for some $\zeta > 0$:

$$n_i(t) \leq \zeta + \sum_{s=\zeta+1}^{t} \mathbb{1}\left(S_i(t) + c\sqrt{\frac{\log t}{n_i(t)}} \geq S_j(t) + c\sqrt{\frac{\log t}{n_j(t)}} - \tau(t)\right)$$

$$\leq \zeta + \sum_{s=\zeta+1}^{t} \mathbb{1}\left(\min_{0 < n^* < s} S_j(n^*) + c\sqrt{\frac{\log s}{n^*}} - \tau(n^*) \leq \max_{\zeta < n < s} S_i(n) + c\sqrt{\frac{\log s}{n}}, j = 1 \text{ or } 2\right)$$

$$\leq \zeta + \sum_{j \in \{1,2\}} \sum_{s=1}^{\infty} \sum_{n^*=1}^{s-1} \sum_{n=\zeta}^{s-1} \mathbb{1}\left(S_j(n^*) + c\sqrt{\frac{\log s}{n^*}} - \tau(s) \leq S_i(n) + c\sqrt{\frac{\log s}{n}}, j = 1 \text{ or } 2\right) \ .$$

Take expectation and set $\zeta = \frac{(2c)^2 \log t}{(\frac{b\underline{L}\Delta}{2})^2}$ we know

$$\mathbb{E}[n_i(T)] \leq \frac{(2c)^2 \log t}{(\frac{b\underline{L}\Delta}{2})^2} + \sum_{j \in \{1,2\}} \sum_{s=1}^{\infty} \sum_{n^*=1}^{s-1} \sum_{n=\zeta}^{s-1} \mathrm{O}(\frac{1}{s^4})$$

$$\leq \frac{(2c)^2 \log t}{(\frac{b\underline{L}\Delta}{2})^2} + \sum_{j \in \{1,2\}} \sum_{s=1}^{\infty} \mathrm{O}(\frac{1}{s^2})$$

$$\leq \frac{(2c)^2 \log t}{(\frac{b\underline{L}\Delta}{2})^2} + \text{const.} = \mathrm{O}(\frac{\log T}{\Delta^2}) \ .$$

$\square$

## 8.8 Proof for Theorem 2

*Proof.* We first prove that regardless of workers' decision on efforts exertion we will be having:

**Lemma 9.** *Under SR-UCB for linear least square, we have when $t$ is large $n_i(t) = \Omega(\log t), a.s.$*

Note the classical bandit argument cannot be applied directly to establish a $O(\log t)$ lower bound since the underlying distribution for the index terms can be different for different arms, as now $S_i(t)$ depends not only on each worker's parameter $e_i$, but will also depend on other workers $e_i$ and their labeled data. With the help of this lemma we have the following results:

**Lemma 10.** *At any time $t$, the number of selection of workers $i > 2$ with $e_i(t) \leq e_1^* + \gamma, \forall t$ satisfies $\mathbb{E}[n_i(t)] = O\left(\frac{\log t}{\Delta^2}\right)$. And moreover if $e_1(t) \equiv e_1^*, e_2(t) \equiv e_2^*$, we will be having $\mathbb{E}[n_1(t)], \mathbb{E}[n_2(t)] = T - O(\log T)$.*

Also since $\sigma_1(e_1^*) = \sigma_2(e_2^*)$, following previous argument for Lemma 6 we can similarly establish that there exists a constant $\delta_U > 0$ s.t. with probability at least $1 - O\left(\frac{1}{t^2}\right)$, $\frac{1}{1+\delta_U} n_2(t) \leq n_1(t) \leq (1 + \delta_U) n_2(t)$, following which we know $\mathbb{E}[n_1(t)], \mathbb{E}[n_2(t)] \geq T - O(\log T)$. Therefore further deviating to $e_{1(2)} > e_{1(2)}^*$ will give the corresponding worker at most $O\left(\frac{\log T}{T}\right) < O\left(\sqrt{\frac{\log T}{T}}\right)$ additional profit per task. For deviation to $e_i < e_i^*, i = 1, 2$, similar to the symmetric case we can again show such a deviation can bring in at most $O\left(\sqrt{\frac{\log T}{T}}\right)$ additional payment: what we need to establish is similar to Lemma 8 that

$$\mathbb{E}[n_2(T; t \geq T^{z'})] \leq O\left(\frac{\log T}{\Delta^2}\right) = O(T^z).$$

With above we establish the fact that exerting efforts $e_1^*, e_2^*$ is $O\left(\sqrt{\frac{\log T}{T}}\right) - \text{BNE}$ for worker 1 & 2.

For worker $i > 2$, since we already proved that for any effort level $e_i \leq e_1^* + \gamma$, the expected number of selection is bounded up by $O(\log T)$, as $\gamma$ is set to be small enough such that $\bar{L}\gamma \leq \frac{\Delta_1}{2}$, and we will then be having $\sigma_i(e_i) - \sigma_1(e_1^*) \geq \frac{\Delta_1}{2}$. Therefore any profitable deviation will lead to at most $O\left(\frac{\log T}{T}\right) < O\left(\sqrt{\frac{\log T}{T}}\right)$ additional profit per task. Apparently deviating to $e_i > e_1^* + \gamma$ is not profitable at all (negative marginal gain).

Again consider the dynamic case, where workers can choose to exert different level of efforts at each different steps. When exerting efforts to reach the same effort level as worker 1 & 2 $\frac{\sum_{t=1}^T \sigma_i(e_i(t))}{T} = \sigma_1(e_1^*)$, [9] suppose $\sigma_1(e_1^*) = \sigma_i(e_i^*)$ and we know $e_i^* > e_1^*$; and further

$$\sigma_i(e_i^*) - \sigma_i(e_1^*) \leq \bar{L}(e_i^* - e_1^*) \Rightarrow e_i^* \geq \frac{\Delta}{\bar{L}} + e_1^*.$$

Also we have (by convexity)

$$\sigma_i(e_i^*) = \frac{\sum_{t=1}^T \sigma_i(e_i(t))}{T} \geq \sigma_i\left(\frac{\sum_{t=1}^T e_i(t)}{T}\right) \Rightarrow \frac{\sum_{t=1}^T e_i(t)}{T} \geq e_i^* \geq \frac{\Delta}{\bar{L}} + e_1^* \geq e_1^* + \gamma$$

However the average payment is only $e_1^* + \gamma$, we know such a deviation is not profitable for workers $i > 2$. ☐

## 8.9 Proof for Lemma 9

*Proof.* Suppose there is $i$, such that $n_i(t) = o(\log t)$. The basic intuition of a contradiction is as follows: denote the worker with maximum number of selection as $j \neq i$, and we know $n_j(t) = O(t)$. Then we will have

$$I_i(t) \geq a - 4M^2 b + c\sqrt{\frac{\log t}{n_i(t)}} > a + c\sqrt{\frac{\log t}{O(t)}} - \tau(t) \geq I_j(t) - \tau(t), \tag{8.12}$$

thus $i$ will be selected when $t$ is large. More rigorously consider $t'$ as the earliest time such that $n_j(t') \geq t^z$. We know $t^z \leq t' \leq t - t^z$, where $0 < z < 1$ is a constant, and the second inequality comes as otherwise $n_i(t) \leq n_j(t') + t^z \leq t^z + t^z + 1 < O(t)$. Then

$$n_i(t) \geq \sum_{n=t'}^{t} \mathbb{1}(j \in d(n)) \cdot \mathbb{1}\left(c\sqrt{\frac{\log n}{n_i(n)}} \geq 4M^2 b + c\sqrt{\frac{\log n}{n_j(n)}}\right)$$

$$\geq \sum_{n=t'}^{t} \mathbb{1}(j \in d(n)) \cdot \mathbb{1}\left(c\sqrt{\frac{z\log t}{n_i(t)}} \geq 4M^2 b + c\sqrt{\frac{\log t}{t^z}}\right),$$

where the second inequality comes from the facts that

$$\frac{\log n}{n_i(n)} \geq \frac{\log t'}{n_i(t)}, \quad \frac{\log n}{n_j(n)} \leq \frac{\log t}{n_j(t')} \leq \frac{\log t}{t^z} \ .$$

If $n_i(t) = o(\log t)$, when $t$ is large

$$\mathbb{1}\left(c\sqrt{\frac{z\log t}{n_i(t)}} \geq 4M^2 b + c\sqrt{\frac{\log t}{t^z}}\right) = 1 \Rightarrow n_i(t) \geq \sum_{n=t'}^{t} \mathbb{1}(j \in d(n)) = O(t),$$

which is a contradiction. □

## 8.10  Proof for Lemma 10

*Proof.* With an appropriately selected $\gamma$, for $i > 2$, worker $i$ will only be willing to pay $e_i \leq e_1^* + \gamma$, as otherwise no matter how many times they got selected, they always receive negative payment. Therefore we will be having $\sigma_i(e_i) \geq \sigma_i(e_1^* + \gamma) \geq \sigma_i(e_1^*) - \bar{L}\gamma$. If we make $\gamma$ small enough such that $\bar{L}\gamma \leq \frac{\Delta_1}{2}$ , we will then be having $\sigma_i(e_i) - \sigma_1(e_1^*) \geq \frac{\Delta_1}{2}$, which leads to $b\frac{\Delta_1}{2}$ difference in the expected value of index. With this, following the proof for Lemma (8), we upper bound the number of selections on the order of $O\left(\frac{\log T}{\Delta_1^2}\right)$: the only difference is in bounding the following event:

$$\{b\frac{\Delta_1}{2} \leq 2c\sqrt{\frac{\log t}{n_i(t)}} + \tau(t) + \frac{C_1}{(\sum_{k \neq i} n_k(t))^2} + \frac{C_1}{(\sum_{k \neq j} n_k(t))^2})\} \ .$$

By Lemma 9, we claim, a.s.,

$$(\sum_{k \neq i} n_k(t))^2 \geq O((\log t)^2), \quad (\sum_{k \neq j} n_k(t))^2 \geq O((\log t)^2),$$

and thus since $\Delta$ is a positive constant, we know when $t$ is large enough, there exists a $\Delta' = \alpha\Delta_1$ where $0 < \alpha < 1$ such that

$$\{b\frac{\Delta_1}{2} \leq 2c\sqrt{\frac{\log t}{n_i(t)}} - \tau(t) - O\left(\frac{1}{(\sum_{k \neq i} n_k(t))^2}\right) - O\left(\frac{1}{(\sum_{k \neq j} n_k(t))^2}\right)\}$$

$$\subseteq \{b\frac{\Delta'}{2} \leq 2c\sqrt{\frac{\log t}{n_i(t)}}\},$$

from where we can follow the reasoning for Lemma (8) to finish the proof. □

## 8.11  Removing bad equilibira

*Proof.* When following the equilibria $e^*$, the average utility for each worker is $p_s\gamma$. With adding this independent randomization device, most of the key parts of the proof will go through. For example, the proof of Lemma 6 and 7 will go through directly, except for the small change that, the number of selections up to time $t$ is now lower bounded (instead of being lower bounded by $t$) by the following random variable that satisfies: denote the event of a selection as $s(t) \in \{1(\text{selection}), 0(\text{no selection})\}$

$$\Pr\left[\frac{\sum_n s(n)}{t} \geq p_s - \sqrt{\frac{\log t}{t}}\right] \leq \exp(-2\frac{\log t}{t} \cdot t) = 1/t^2.$$

Then based on Lemma 7, we know with $O(T)$ number of times, the worker will be jointly selected with others. This finishes the $p_s \cdot \gamma$ argument.

Now if worker $i$ deviates, his utility will be upper bounded by the following case (1) he becomes the monopoly for $O(T)$ number of times. (2) His marginal gain is upper bounded by $\gamma + O(\sqrt{\frac{\log T}{T}})$. (as otherwise if the deviation of effort is higher than $O(\sqrt{\frac{\log T}{T}})$, the number of selection will be upper bounded at the order of sublinear). Then the utility gain for such a deviation is

$$\gamma + O(\sqrt{\frac{\log T}{T}}) - p_s \cdot \gamma = O(\sqrt{\frac{\log T}{T}}/\gamma) \cdot \gamma + O(\sqrt{\frac{\log T}{T}}) = O(\sqrt{\frac{\log T}{T}}).$$

This establishes the $O(\sqrt{\frac{\log T}{T}})$-BNE.

When others are exerting $e = e^* - \Delta e$ ($\Delta e > O(\sqrt{\frac{\log T}{T}})$). If a particular worker $i$ is also exerting the same level of effort, the average payoff is $p_s(\gamma + \Delta e)$. However if the worker deviates by exerting $e = e^* - \Delta e + \tilde{\Delta} e$, where $\tilde{\Delta} e > O(\sqrt{\frac{\log T}{T}})$, we have the number of times the other workers being selected bounded by (by Lemma 8): $O(\frac{\log T}{\tilde{\Delta} e^2}) = o(T)$. This fact helps establish that the number of unique selection for worker $i$ is $O(T)$. Then his marginal payment will become $\gamma + \Delta e - \tilde{\Delta} e$. The gain of such a deviation is

$$O(\frac{\sqrt{\log T/T}}{\gamma})(\gamma + \Delta e - \tilde{\Delta} e) > O(\sqrt{\frac{\log T}{T}})$$

when $\Delta e > O(\gamma)$ and $\tilde{\Delta} e < \sqrt{\log T/T} \cdot \frac{\Delta e}{\gamma}$. □

## 8.12 With unknown σ

*Proof.* Within our sequential learning setting, we now show we can even afford to assign tasks and induce efforts without knowing the exact σ values. The idea is as follows: we fix an arbitrary effort level for a certain period of time, and at any time $t$, we can use collected data from past with this particular effort level to learn a regression model $\theta(t)$. Using this estimated regression model, we are able to estimate $\sigma(e)$. When the space of effort level is finite, we can further impose a bandit selection procedure over effort(i.e., the effort levels are bandits). When the effort level is continuous, using the assumption we made earlier that σ is continuous in $e$, and suppose the effort level has a bounded support $[0, \bar{e}]$, we can then separate $[0, \bar{e}]$ into $T^z, 0 < z < 1$ intervals uniformly, with each interval having length $1/T^z$. For each of the interval we assign $T^\kappa$ number of data. Both $0 < z, \kappa < 1$ are constant parameters by design. We choose that $z + \kappa < 1$. For each interval $k = 1, ..., T^z$, we assign $e(k) = \frac{k}{T^z}\bar{e}$. Then use the $T^\kappa$ samples to estimate $\sigma(e(k))$ in the following way (denote the samples assigned as $(x(n), \tilde{y}(x(n)))$):

$$\tilde{\sigma}(e(k)) = \frac{\sum_{n=1}^{\kappa}(\theta^T(T^\kappa)x(n) - \tilde{y}(x(n)))^2}{T^\kappa} .$$

Now we have the following lemma,

**Lemma 11.** *With SR-UCB for linear least square, but adaptive effort selection mechanism detailed above, with probability at least $1 - O(\frac{1}{T^2})$, $|\tilde{\sigma}(e) - \sigma(e)| \le O(\sqrt{\frac{\kappa \log T}{T^\kappa}}) + O(\frac{1}{T^z}), \forall e.$*

*Proof.* Using Chernoff bound, we can prove the following concentration results for the estimation when $T^\kappa$ samples are available: with probability at least $1 - O(\frac{1}{T^2})$,

$$|\tilde{\sigma}(e(k)) - \sigma(e(k))| \le O(\sqrt{\frac{\kappa \log T}{T^\kappa}}) .$$

Then using Lipschitz condition we know

$$|\tilde{\sigma}(e) - \sigma(e)| \le |\tilde{\sigma}(e) - \sigma(e(k))| + |\sigma(e(k)) - \sigma(e)| \le O(\sqrt{\frac{\kappa \log T}{T^\kappa}}) + O(\frac{1}{T^z}) .$$

Note in order to use such an estimation, we need to make sure that during each interval each worker will exert $e(k)$. We can similarly establish a $O\left(\sqrt{\frac{\log T}{T^\kappa}}\right)$-BNE for worker to contribute the corresponding effort for each interval. The reason that we can decouple the above argument for each interval that worker will exert effort $e(k)$ is due to the fact that net payment $p_i - e_i = \gamma$ is independent of the effort level, so the workers have no incentives to mislead the learner into believing a wrong mapping between $\sigma$ and $e$; and within each assignment block, workers again try to maximize total payment. Therefore for any $e$, suppose $\frac{k-1}{T^z} \le e \le \frac{k}{T^z}$, use $\tilde{\sigma}(e(k))$ to serve as an approximation we will have

$$|\tilde{\sigma}(e) - \sigma(e)| \le |\tilde{\sigma}(e) - \sigma(e(k))| + |\sigma(e(k)) - \sigma(e)| \le O\left(\sqrt{\frac{\kappa \log T}{T^\kappa}}\right) + O\left(\frac{1}{T^z}\right).$$

$\square$

The above error bound reaches the optimal order when $\kappa/2 = z$, and since $\kappa + z < 1$, we have the error decays roughly on the order of $O\left(T^{-1/3}\right)$.

$\square$

## 8.13 Performance with contributed data

*Proof.* For outputting the final regression model, we will use the data from only the top 2 most selected workers. First since $\mathbb{E}[n_i(T)] \ge T - O(\log T)$ for $i = 1, 2$ we know

$$\Pr[T - n_i(T) \ge T/2] \le \frac{\mathbb{E}[T - n_i(T)]}{T/2} \le O\left(\frac{\log T}{T}\right), i = 1, 2.$$

So w.h.p., $n_1(T), n_2(T) \ge T/2$, and then w.p. being at least $1 - e^{-O(T)}$ we know the square error loss of the trained regression model is bounded as follows:

$$\mathbb{E}[\sigma_1(e_1^*)/(\sum_{i=1,2} n_i(T))^2 - \sigma_1(e_1^*)/(2T)^2]$$

$$\le \mathbb{E}\left[\max_{\sum_{i=1,2} n_i(T)} \frac{2\sigma_1(e_1^*)}{(\sum_{i=1,2} n_i(T))^3}(2T - \sum_{i=1,2} n_i(T))\right]$$

$$\le \frac{2\sigma_1(e_1^*)}{T^3}(2T - \mathbb{E}[\sum_{i=1,2} n_i(T)])$$

$$\le \frac{2\sigma_1(e_1^*)}{T^3}(2T - 2T + 2O(\log T))$$

$$= O\left(\frac{\sigma_1(e_1^*) \log T}{T^3}\right),$$

where the first inequality is due to mean value theorem, and the second is due to $\sum_i n_i(T) \ge T$, as there is at least one selection at a time. $\square$

## 8.14 Ridge regression: Proof for Lemma 1

*Proof.* Again denote the stacked data in a matrix form as $\mathbf{X} \in \mathbb{R}^{n \times d}$, and the corresponding labeling outcome $\tilde{y} \in \mathbb{R}^n$. Following classical results from linear regression we know

$$||\tilde{\theta}_{-i}(t) - \mathbb{E}[\tilde{\theta}_{-i}(t)]||_2^2 = \text{trace}((\rho I + \mathbf{X}^T \mathbf{X})^{-1} \mathbf{X}^T (\tilde{y} - y)(\tilde{y} - y)^T \mathbf{X}(\rho I + \mathbf{X}^T \mathbf{X})^{-1}).$$

$$||\mathbb{E}[\tilde{\theta}_{-i}(t)] - \theta||_2^2 = || - \rho(\rho I + \mathbf{X}^T \mathbf{X})^{-1}\theta||_2^2.$$

The variance term is independent of the ground-truth regression model $\theta$ and will converge similarly with our previous arguments. The bias term $||\mathbb{E}[\tilde{\theta}_{-i}(t)] - \theta||_2^2$ is depending on $\theta$ which is unknown. Therefore without knowing such a quantity[10], it is hard for both the workers and requester to evaluate the one step payment rule. Within our dynamic setting, workers do not need to calculate the specific form of $\theta$; instead workers only need to form the belief that under the same effort level, they will have comparable indexes. Further with more and more data being collected, the bias term will

be decreasing and its effects will diminish – this is by observing the following fact that [4] w.h.p. $\geq 1 - e^{-C_1 n}$, when there is $n$ sample being available (following the notations in Lemma 5)

$$||\rho I + \mathbf{X}^T\mathbf{X}|| \leq \rho + (1+\xi)\frac{n}{d+2}, \ ||(\rho I + \mathbf{X}^T\mathbf{X})^{-1}|| \leq \frac{1}{\rho + (1-\xi)\frac{n}{d+2}} \ .$$

As in the proof for Lemma 5, we also know

$$||\mathbf{X}^T(\tilde{y}-y)(\tilde{y}-y)^T\mathbf{X}||_2^2 \leq Z^2\frac{1+\xi}{2+d}n \ ,$$

with which we will be able to prove

$$\text{trace}\left( (\rho I + \mathbf{X}^T\mathbf{X})^{-1}\mathbf{X}^T(\tilde{y}-y)(\tilde{y}-y)^T\mathbf{X}(\rho I + \mathbf{X}^T\mathbf{X})^{-1} \right)$$

$$\leq \left( (\frac{(1+\xi)\frac{n}{d+2}}{(\rho+(1-\xi)\frac{n}{d+2})^2})^2 \cdot Z^2\frac{1+\xi}{2+d}n \right)$$

$$\leq \left( \frac{Z^2(1+\xi)^3(d+2)}{(1-\xi)^4} \right)^2 \frac{1}{n^2} \ ,$$

and

$$||-\rho(\rho I + \mathbf{X}^T\mathbf{X})^{-1}\theta||_2^2 \leq \left( \frac{\rho M}{\rho+(1-\xi)\frac{n}{d+2}} \right)^2$$

$$= (\frac{\rho M(d+2)}{1-\xi})^2 \cdot \left( \frac{1}{\rho(d+2)/(1-\xi)+n} \right)^2$$

$$\leq (\frac{\rho M(d+2)}{1-\xi})^2 \frac{1}{n^2} \ .$$

To summarize

$$||\tilde{\theta}_{-i}(t) - \theta||_2^2 \leq ((\frac{Z^2(1+\xi)^3(d+2)}{(1-\xi)^4})^2 + (\frac{\rho M(d+2)}{1-\xi})^2)\frac{1}{n^2} \ .$$

$$\square$$

### 8.15 (Sketch)-Proof for $\pi$-BNE for Non-linear estimator

*Proof.* Again we evaluate the score for each worker:

$$(\tilde{f}_{-i,t}(x) - \tilde{y}(x))^2 = (\tilde{f}_{-i,t}(x) - y(x))^2 - 2(\tilde{f}_{-i,t}(x) - y(x)) \cdot (z + z_i(e_i)) + (z + z_i(e-i))^2 \ .$$

We want to bound $(\tilde{f}_{-i,t}(x) - y(x))^2$. More specifically according to the results from [20], for non-linear regression model we can establish:

**Lemma 12.** *With $n$ i.i.d. samples, w.h.p.* $||\tilde{\theta}_i(t) - \theta||_2 \leq O(\frac{1}{\sqrt{n}}) \ .$

The key difference that is going to affect establishing the $\pi$-BNE is in proving Lemma 6. With Lipschitz condition we know now w.h.p. $|S_i^1(t)| \leq \frac{C_1}{n}$, where $C_1$ is re-defined as the constant proved in Lemma 12. Again we know w.h.p., $n \geq C_2 t$ which gives us $|S_i^1(t)| \leq \frac{C_1}{C_2 t}$. So what we need to prove is to bound for each $k = 2,3,4,5$

$$\Pr\left[ S_i^k(t) - \mathbb{E}[S_i^k(t)] \geq \frac{\sqrt{1+\delta_U}-1}{8}c\sqrt{\frac{\log t}{n_i(t)}} - \frac{C_1}{4C_2 t} \right] \ .$$

Take $S_i^2(t)$ for example, and the rest follow the same logic.

$$\Pr\left[S_i^2(t) - \mathbb{E}[S_i^2(t)] \geq \frac{\sqrt{1+\delta_U}-1}{8}c\sqrt{\frac{\log t}{n_i(t)}} - \frac{C_1}{4C_2t}\right]$$

$$\leq \exp\left(-2(\frac{\sqrt{1+\delta_U}-1}{8}c\sqrt{\frac{\log t}{n_i(t)}} - \frac{C_1}{4C_2t})^2 n_i(t)\right)$$

$$\leq \exp\left(-\frac{2(\frac{\sqrt{1+\delta_U}-1}{8}c\sqrt{\frac{\log t}{n_i(t)}} - \frac{C_1}{4C_2^2t})^2 n_i(t)}{16b^2M^2Z^2}\right)$$

$$\leq \exp\left(-2\frac{(\frac{\sqrt{1+\delta_U}-1}{8})^2c^2}{16b^2M^2Z^2}\log t\right)\cdot\exp\left(2\frac{\frac{\sqrt{1+\delta_U}-1}{8}c\cdot\frac{C_1}{4C_2^2t}\sqrt{\log t\cdot n_i(t)}}{16b^2M^2Z^2}\right)$$

$$\leq \exp\left(-2\frac{(\frac{\sqrt{1+\delta_U}-1}{8})^2c^2}{16b^2M^2Z^2}\log t\right)\cdot\exp\left(2\frac{\frac{\sqrt{1+\delta_U}-1}{8}c\cdot\frac{C_1}{4C_2^2}}{16b^2M^2Z^2}\right) \leq \frac{\exp(2)}{t^2}\,.$$

$\square$

### 8.16  Example: logistic regression

*Proof.* To see this, denote $\mu := \theta^T x$ and $\tilde{\mu} := \tilde{\theta}_i^T(t)x$ and apply mean value theorem to $\frac{1}{1+e^{-y}}$ we have

$$|\frac{1}{1+e^{-\mu}} - \frac{1}{1+e^{-\tilde{\mu}}}|$$

$$\leq \max(\frac{1}{1+e^{-y}})'|\mu - \tilde{\mu}|$$

$$= \max\frac{1}{e^y + e^{-y} + 2}|(\theta - \tilde{\theta}_i(t))^T x|$$

$$\leq \frac{1}{4}|(\theta - \tilde{\theta}_i(t))^T x|\,,$$

where we used the fact $e^y + e^{-y} \geq 2$. Since

$$|(\theta - \tilde{\theta}_i(t))^T x| \leq ||\theta - \tilde{\theta}_i(t)||_2 ||x||_2 \leq ||\theta - \tilde{\theta}_i(t)||_2,$$

we proved the claim. $\square$

## 9  Proofs for Section 6

### 9.1  $O\left(\sqrt{\log T/T}\right)$-BNE for OSR1-UCB

*Proof.* Following the results detailed in [19] for online learning algorithm for strongly convex function ($\rho$-ridge regularized loss function is $2\rho$-strongly convex), set $\eta_t = 1/(2\rho t)$ we have

**Lemma 13.** $\forall t$ of OSR1-UCB, w.p. $\geq 1 - \delta$, $||\tilde{\theta}_{-i}^{online}(t) - \tilde{\theta}_{-i}(t)||_2^2 \leq O\left(\log(\log t/\delta)/\rho t\right)$.

We can similarly establish the $O\left(\sqrt{\log T/T}\right)$-BNE for effort exertion – the only difference is compared to what we established earlier that $||\tilde{\theta}_{-i}(t) - \theta||_2^2 \leq O\left(1/t^2\right)$, here we will have a $O\left(\log t/\rho t\right)$ (by setting $\delta = 1/t^2$) convergence rate which is much slower in the order. Nevertheless we show this is enough – the intuition is $O\left(\log t/\rho t\right) < O\left(\sqrt{\log t/t}\right)$ which is the order of the bias term in our SR-UCB index, such small converging term will not affect the analysis by much. The argument is similar to the proof in Section 8.15, in that we only need to prove bound on $S_i^k(t) - \mathbb{E}[S_i^k(t)], k = 1, 2, ..., 5$

with a different confidence/bias term. Take $S_i^2(t)$ for example:

$$\Pr\left[S_i^2(t) - \mathbb{E}[S_i^2(t)] \geq \frac{\sqrt{1+\delta_U}-1}{8}c\sqrt{\frac{\log t}{n_i(t)}} - \frac{C_1\log t}{4C_2 t}\right]$$

$$\leq \exp\left(-2\left(\frac{\sqrt{1+\delta_U}-1}{8}c\sqrt{\frac{\log t}{n_i(t)}} - \frac{C_1\log t}{4C_2 t}\right)^2 n_i(t)\right)$$

$$\leq \exp\left(-\frac{2\left(\frac{\sqrt{1+\delta_U}-1}{8}c\sqrt{\frac{\log t}{n_i(t)}} - \frac{C_1\log t}{4C_2 t}\right)^2 n_i(t)}{16b^2 M^2 Z^2}\right)$$

$$\leq \exp\left(-2\frac{(\frac{\sqrt{1+\delta_U}-1}{8})^2 c^2}{16b^2 M^2 Z^2}\log t\right)\cdot\exp\left(2\frac{\frac{\sqrt{1+\delta_U}-1}{8}c\cdot\frac{C_1\log t}{4C_2 t}\sqrt{\log t \cdot n_i(t)}}{16b^2 M^2 Z^2}\right)$$

$$\leq \exp\left(-2\frac{(\frac{\sqrt{1+\delta_U}-1}{8})^2 c^2}{16b^2 M^2 Z^2}\log t\right)\cdot\exp\left(2\frac{\frac{\sqrt{1+\delta_U}-1}{8}c\cdot\frac{C_1}{4C_2^2}\log t\sqrt{\frac{\log t}{t}}}{16b^2 M^2 Z^2}\right).$$

If we choose $\frac{\sqrt{1+\delta_U}-1}{8}c \geq 4bMZ\cdot\max\{1, \frac{4C_2^2}{C_1}\}$, and $t \geq 100$, we know

$$\exp\left(-2\frac{(\frac{\sqrt{1+\delta_U}-1}{8})^2 c^2}{16b^2 M^2 Z^2}\log t\right)\cdot\exp\left(2\frac{\frac{\sqrt{1+\delta_U}-1}{8}c\cdot\frac{C_1}{4C_2^2}\log t\sqrt{\frac{\log t}{t}}}{16b^2 M^2 Z^2}\right)$$

$$\leq \exp(-2\log t)\exp(2) \leq \frac{\exp(2)}{t^2}.$$

$\square$

## 9.2 $\mathrm{O}\left(\log T/\sqrt{T}\right)$-BNE for OSR2-UCB

*Proof.* First we prove

**Lemma 14.** *With $S_i^{online}(t)$, $\forall t$, w.p. $1 - \mathrm{O}\left(1/t^2\right)$:*

$$\frac{1}{t}\sum_{n=1}^t \mathbb{1}(i \in d(n))\left(\left(\tilde{\theta}_{-i}^{online}(n) - \theta\right)^T x_i(n)\right)^2 \leq \mathrm{O}\left(\log t/\sqrt{n_i(t)}\right).$$

Then the argument is similar to the proof in Section 8.15, in that we again need to prove bound on $S_i^k(t) - \mathbb{E}[S_i^k(t)], k = 1, 2, ..., 5$ with a different confidence/bias bound. Take $S_i^2(t)$ for example:

$$\Pr\left[S_i^2(t) - \mathbb{E}[S_i^2(t)] \geq \frac{\sqrt{1+\delta_U}-1}{8}c\sqrt{\frac{\log t}{n_i(t)}} - \frac{C_1\log t}{4C_2\sqrt{n_i(t)}}\right]$$

$$\leq \exp\left(-2\left(\frac{\sqrt{1+\delta_U}-1}{8}c\sqrt{\frac{\log^2 t}{n_i(t)}} - \frac{C_1\log t}{4C_2\sqrt{n_i(t)}}\right)^2 n_i(t)\right)$$

$$\leq \exp\left(-\frac{2\left(\frac{\sqrt{1+\delta_U}-1}{8}c\sqrt{\frac{\log^2 t}{n_i(t)}} - \frac{C_1\log t}{4C_2\sqrt{n_i(t)}}\right)^2 n_i(t)}{16b^2 M^2 Z^2}\right)$$

$$\leq \exp\left(-2\frac{(\frac{\sqrt{1+\delta_U}-1}{8})^2 c^2}{16b^2 M^2 Z^2}\log^2 t\right)\cdot\exp\left(2\frac{\frac{\sqrt{1+\delta_U}-1}{8}c\cdot\frac{C_1\log t}{4C_2\sqrt{n_i(t)}}\log t\sqrt{n_i(t)}}{16b^2 M^2 Z^2}\right)$$

$$\leq \exp\left(-2\frac{(\frac{\sqrt{1+\delta_U}-1}{8})^2 c^2}{16b^2 M^2 Z^2}\log^2 t\right)\cdot\exp\left(2\frac{\frac{\sqrt{1+\delta_U}-1}{8}c\cdot\frac{C_1}{4C_2^2}\log^2 t}{16b^2 M^2 Z^2}\right).$$

If we choose $\frac{\sqrt{1+\delta_U}-1}{8}c \geq \max\{2\frac{C_1}{4C_2^2}, 4bMZ\}$ , we know

$$\exp\left(-2\frac{(\frac{\sqrt{1+\delta_U}-1}{8})^2 c^2}{16b^2M^2Z^2}\log^2 t\right)\cdot\exp\left(2\frac{\frac{\sqrt{1+\delta_U}-1}{8}c\cdot\frac{C_1}{4C_2^2}\log^2 t}{16b^2M^2Z^2}\right) \leq \exp(-\log^2 t) \leq \frac{1}{t^2} \text{ , if } t \geq e^2.$$

$\square$

### 9.3 Proof for Lemma 14

*Proof.* As in Lemma 13, let $\delta = \frac{1}{t^3}$, we know with probability at least $1-\frac{1}{t^3}$ we have

$$((\tilde{\theta}_{-i}^{\text{online}}(n) - \theta)^T x_i(n))^2 \leq ||\tilde{\theta}_{-i}^{\text{online}}(n) - \theta||_2^2 \leq O(\frac{\log t}{n}) \ .$$

Via union bounds with probability at least $1-\frac{1}{t^2}$,

$$\sum_{n=1}^{n_i(t)}((\tilde{\theta}_{-i}^{\text{online}}(n) - \theta)^T x_i(n))^2 = \sqrt{n_i(t)}2M^2 + \sum_{n=\sqrt{n_i(t)}}^{n_i(t)} O(\frac{\log t}{n}) = O(\log t \log n_i(t)) \ .$$

which leads to the average error

$$\frac{\sqrt{n_i(t)}2M^2 + \log t \log n_i(t)}{n_i(t)} \leq \frac{\log t}{\sqrt{n_i(t)}} \ ,$$

which is due to the fact shown in Lemma 9 that when $t$ is large, $n_i(t) \geq O(\log t)$ a.s. and when $n_i(t) = \Omega(\log t)$ we have $\sqrt{n_i(t)} \geq \log n_i(t)$ when $t$ is large. $\square$

## 10 Proof for Section 7

### 10.1 Proof for Lemma 2

*Proof.* This can be proved by induction. At time $t = 2$, $n_i(2)$ is a function of $\{S_j(1)\}_j$, the initial value. Assume this is true for $t$. Consider time $t + 1$. $n_i(t + 1)$ is an outcome from an ordering function based on inputs of $\{S_j(t)\}_j$ and $\{n_j(t)\}_j$. Based on induction hypothesis, $\{n_j(t)\}_j$ can be written as functions of $\{S_j(n), n < t\}_j$. With this we proved that $\{n_j(t+1)\}_j$ can also be written as functions of $\{S_j(n), n < t+1\}_j$. Proved. $\square$

### 10.2 Proof for Lemma 3

*Proof.* We first prove that a finite deviation from worker $i$ creates at most $O(1/T)$ differences in $\tilde{\theta}(T)$ (sensitivity) with high probability. Denote this event as $\mathcal{E}(T)$, we know $\Pr[\mathcal{E}(T)] \leq e^{-KT}$. Shorthand the contributed data as $\tilde{y}_i(n) := \tilde{y}_i(n, e_i(n))$, $\tilde{y}'_i(n) := \tilde{y}'_i(n, e'_i(n))$. And denote the regression model trained with $\tilde{y}_i(n), \tilde{y}'_i(n)$ as $\tilde{\theta}(T), \tilde{\theta}'(T)$ (differ only in one data point). Then we have

$$\Pr[\tilde{\theta}^p(T)|\tilde{y}'_i(n)] = \Pr[\tilde{\theta}^p(T)|\tilde{y}'_i(n), \overline{\mathcal{E}(T)}] \cdot \Pr[\overline{\mathcal{E}(T)}] + \Pr[\tilde{\theta}^p(T)|\tilde{y}'_i(n), \mathcal{E}(T)]\Pr[\mathcal{E}(T)]$$
$$\leq \Pr[\tilde{\theta}^p(T)|\tilde{y}'_i(n), \overline{\mathcal{E}(T)}] + O(e^{-KT}) \ .$$

Consider the first term above $\Pr[\tilde{\theta}^p(T)|\tilde{y}'_i(n), \overline{\mathcal{E}(T)}]$:

$$\Pr[\tilde{\theta}^p(T)|\tilde{y}'_i(n), \overline{\mathcal{E}(T)}] = \Pr[v_\theta = \tilde{\theta}^p(T) - \tilde{\theta}'(T)]$$
$$= \Pr[v_\theta = \tilde{\theta}^p(T) - \tilde{\theta}(T) + \tilde{\theta}'(T) - \tilde{\theta}(T)]$$
$$= C \cdot \exp(-\varepsilon_\theta||\tilde{\theta}^p(T) - \tilde{\theta}(T) + \tilde{\theta}'(T) - \tilde{\theta}(T)||_2)$$
$$\leq C \cdot \exp(-\varepsilon_\theta||\tilde{\theta}^p(T) - \tilde{\theta}(T)||_2) \cdot \exp(\varepsilon_\theta||\tilde{\theta}'(T) - \tilde{\theta}(T)||_2)$$
$$= \Pr[\tilde{\theta}^p(T)|\tilde{y}_i(n)] \cdot \exp(\varepsilon_\theta \cdot O(\frac{1}{T})) \ .$$

What is left to prove is that $\tilde{\theta}(T)$'s sensitivity is $O(\frac{1}{T})$ w.h.p., that is we want to bound the difference in the regression model: $||\tilde{\theta}(T) - \tilde{\theta}'(T)||_2$. First

$$||\tilde{\theta}(T) - \tilde{\theta}'(T)||_2$$
$$= ||\tilde{\theta}(T) - \theta + \theta - \tilde{\theta}'(T)||_2$$
$$\leq ||\tilde{\theta}(T) - \theta||_2 + ||\theta - \tilde{\theta}'(T)||_2 .$$

Since $\sum_i n_i(T) \geq T$, by results from Theorem 4 we know with probability at least $1 - e^{-KT}$,

$$||\tilde{\theta}(T) - \theta||_2 \leq O(\frac{1}{T}), \ ||\theta - \tilde{\theta}'(T)||_2 \leq O(\frac{1}{T}) .$$

Thus we know (via union bound) w.p. being at leat $1 - 2e^{-KT}$,

$$||\tilde{\theta}(T) - \tilde{\theta}'(T)||_2 \leq O(\frac{1}{T}) .$$

Also notice that by the CDF of Laplacian distribution,

$$\Pr\left[||\mathbf{v}_\theta||_2 \geq \frac{\log T}{\sqrt{T}}\right] = \exp(-\varepsilon_\theta \cdot \frac{\log T}{\sqrt{T}}) = \exp(-2\sqrt{T}\frac{\log T}{\sqrt{T}}) = \frac{1}{T^2} .$$

$\square$

## 10.3 Proof for Theorem 3

*Proof.* First for $S_i^{\text{online}}(t)$, each $\tilde{y}_i(\cdot)$ appears in at most $\log T + 1$ partial sums. The reason is that a noisy partial sum is discarded only when the size of the partial sum doubles (combine two partial sums). So if the number of such partial sum is greater than $\log T + 1$ we will have the total number of data being greater than $2^{\log T} = T$ which is a contradiction. Then by composition theory we know the privacy leakage in $S_i^{\text{online}}(t)$ is bounded by $O((\log T + 1)\varepsilon) = O(\frac{1}{\log^2 T})$.

Now consider the privacy leakage in $\tilde{\theta}_{-j}^{\text{online}}(t)$. First we prove the following:

**Lemma 15.** *The sensitivity of $\tilde{\theta}_{-j}(t)$ for each $\tilde{y}_i(n, e_i(n)), n \leq t, i \neq j$ is $||\tilde{\theta}_{-j}(t) - \tilde{\theta}'_{-j}(t)||_2 \leq O(1/t)$ with probability at least $1 - O(1/t^3)$.*

The reasoning is similar to a combination of proof for Lemma 7 and Lemma 3. First similar to Lemma 7, we can prove the number of samples that come from $j \neq i$ is at the order of $O(t)$ with probability at least $1 - O(1/t^3)$. Then with $O(t)$ samples, similar to Lemma 3, $||\tilde{\theta}_{-j}(t) - \tilde{\theta}'_{-j}(t)||_2 \leq O(1/t)$ with probability at least $1 - e^{-O(t)}$. Combine above we proved the Lemma.

Again due to the decoupling procedure of the partial sum, each $\tilde{\theta}_{-j}(t)$ appears in at most $\log T + 1$ partial sums ( [3]). Then the privacy leakage of $\tilde{y}_i(n, e_i(n))$ from $\tilde{\theta}_{-j}(t)$ is bounded as $(\log T + 1)\varepsilon \cdot O(\frac{1}{t})$, with probability at least $1 - O(1/t^3)$ (similar to the argument made in the proof for Section 7). Based on this we know for $t \geq O(\log T)$, with an appropriately selected constant we know w.p. at least $1 - O(\frac{1}{\log T} \cdot \frac{1}{t^2})$, we have the sensitivity of $\tilde{\theta}_{-j}(t)$ is at the order of $O(1/t)$. Via union bound we know with probability at least

$$1 - \sum_t O\left(\frac{1}{\log T} \cdot \frac{1}{t^2}\right) = 1 - O\left(\frac{1}{\log T}\right),$$

it satisfies for all $t \geq O(\log T)$, the sensitivity results hold. Sum over all $\tilde{\theta}_{-j}(t)$, we have by composition theory the total privacy leakage is

$$\sum_{t=1}^{O(\log T)} (\log T + 1)\varepsilon \cdot O(1) + \sum_{t=O(\log T)}^{T} (\log T + 1)\varepsilon \cdot O\left(\frac{1}{t}\right) = O((\log T)^2) \cdot \varepsilon .$$

Then we set $\varepsilon = \frac{1}{\log^4 T}$ we have the preserved privacy level is at the order of $O(\frac{1}{\log T})$. Combined with Lemma 3, composited with the privacy preserving level in $\tilde{\theta}^p(T)$ we proved the theorem.

$\square$

## 10.4  $\mathrm{O}\big(\log^6 T/\sqrt{T}\big)$-BNE for PSR-UCB

**Theorem 5.** *With PSR-UCB for linear regression, set fixed payment $p_i$ for all workers as follows: $p_i = e^* + \gamma$, $\gamma = \Omega(\log^6 T/\sqrt{T})$ , and set $c$ to be large enough $c \geq Const.(M,Z,N,b)$. Then exerting effort $e^*$ is $\mathrm{O}\big(\log^6 T/\sqrt{T}\big)$-BNE.*

*Proof.* The main challenge of the proof is to re-establish the convergence of indexes $I_i(t)$ with newly added noises, so that the noise exertion process will not make the indexes useless. There are three sources of noises:

- 1. Noise in $\tilde{\theta}_{-i}^{\mathrm{online}}(t)$, due to the change to a batch summation: $\tilde{\tilde{\theta}}_{-j}^{\mathrm{online}}(t) := \sum_{n=1}^{t} \tilde{\theta}_{-j}(n)/t$.

- 2. Noise in $S_i^{\mathrm{online}}(t)$, due to added noise $v_S$ to partial sums for privacy preserving.

- 3. Noise in $\tilde{\tilde{\theta}}_{-i}^{\mathrm{online}}(t)$, due to added noise $\mathbf{v}_{\tilde{\theta}}$ to partial sums for privacy preserving.

1. First of all we show with the averaging $\tilde{\theta}_{-i}^{\mathrm{online}}(t)$ we do not loss too much performance in converging. Denote $n(t)$ as the number of updates on $\tilde{\theta}_{-j}^{\mathrm{online}}(t)$ up to time $t$. Notice

$$||\tilde{\theta}_{-j}^{\mathrm{online}}(t) - \theta||_2 = ||\frac{\sum_{n=1}^{n_i(t)} \tilde{\theta}_{-j}(n)}{t} - \theta||_2 \leq \frac{\sum_{n=1}^{n_i(t)} ||\tilde{\theta}_{-j}(n) - \theta||_2}{t} .$$

Consider the summation from $n = 1$ to $n_i(t)$. Select a constant $D$. For $n < D\sqrt{n_i(t)}$, we have

$$\sum_{n=1}^{D\sqrt{n_i(t)}} ((\tilde{\theta}_{-i}^{\mathrm{online}}(n) - \theta)^T x_i(n))^2 \leq 2M^2 D\sqrt{n_i(t)} .$$

For $n \geq D\sqrt{n_i(t)}$, since we know $n_i(t) \geq (\log T)^6 \log^6 t$ a.s. (similarly argued as in Lemma 9, but with different bias order), we know $\sqrt{n_i(t)} \geq \mathrm{O}(\log T), a.s.$ Therefore for such $n$, with probability at least $\frac{1}{T^3}$ we will be having

$$||\tilde{\theta}_{-j}^{\mathrm{online}}(n) - \theta||_2 \leq \mathrm{O}\big(\frac{1}{\sqrt{n}}\big). \tag{10.1}$$

And sum over

$$\sum_{n \geq D\sqrt{n_i(t)}} \mathrm{O}\big(\frac{1}{\sqrt{n}}\big) = \mathrm{O}\big(\sqrt{n_i(t)}\big).$$

Using union bound we have w.p. being at least $1 - \frac{1}{T^2}$

$$\frac{2M^2 D\sqrt{n_i(t)} + \mathrm{O}\big(\sqrt{n_i(t)}\big)}{n_i(t)} = \mathrm{O}\big(\frac{1}{\sqrt{n_i(t)}}\big) .$$

2. Now we analyze how these noises affect the accuracy of our indexes. First consider the added noise in $\sum (\tilde{\theta}_i^T(n) x_i(n) - \tilde{y}_i(n))^2$ (as in $S_i^{\mathrm{online}}(t)$). At any time $t$ we have added at most $\lceil \log t \rceil$ number of Laplacian noise with parameter $\varepsilon$. Denote the sum of $E(t) := \sum_{k=1}^{\lceil \log t \rceil} v_S(k)$. From Lemma 2.8 in [3], we know

$$\Pr[|E(t)| > \lambda] \leq 2\exp(-\frac{\lambda^2}{8\lceil \log t \rceil \frac{1}{\varepsilon^2}}) \leq \exp\left(-\frac{\lambda^2}{8} \frac{1}{(\log t + 1)(\log T + 1)^6}\right) . \tag{10.2}$$

Let $\lambda := 4(\log t + 1)(\log T + 1)^3$ we have

$$\Pr\left[|E(t)| > 4(\log t + 1)(\log T + 1)^3\right]$$

$$\leq 2\exp\left(-\frac{16(\log t + 1)^2(\log T + 1)^6}{8} \frac{1}{(\log t + 1)(\log T + 1)^6}\right)$$

$$\leq 2\exp(-2\log t) = 2/t^2 .$$

Note this additional error term is creating a larger than the index bias by order: $O\left(\frac{(\log t + 1)(\log T + 1)^3}{n_i(t)}\right)$.

3. Now consider the noise $\mathbf{v}_{\tilde{\theta}}$ inserted in $\tilde{\theta}_{-i}^{\text{online}}(t)$. Denote $\mathbf{E}(t) := \sum_{k=1}^{\lceil \log t \rceil} \mathbf{v}_{\tilde{\theta}}(k)$. Consider the following fact: for any sample $(\mathbf{x}, y)$

$$((\theta + \mathbf{E}(t)/t)^T \cdot \mathbf{x} - y)^2 = (\theta^T \cdot \mathbf{x} - y)^2 + (\mathbf{E}^T(t)/t \cdot \mathbf{x})^2 + 2\mathbf{E}^T(t)/t \cdot \mathbf{x} \cdot (\theta^T \cdot \mathbf{x} - y)$$

The additional noises appear in two terms:

$$|2\mathbf{E}^T(t)\mathbf{x} \cdot (\theta^T \cdot \mathbf{x} - y)| \leq O(\|\mathbf{E}^T(t)\|_2),$$

due to boundedness of $\mathbf{x}$ and $\theta^T \cdot \mathbf{x} - y$. For the other quadratic term: $(\mathbf{E}^T(t)\mathbf{x})^2 \leq \|\mathbf{E}(t)\|_2^2$. For $\|\mathbf{E}(t)\|_2^2$ we know

$$\|\mathbf{E}(t)\|_2^2 \leq \left(\sum_{k=1}^{\lceil \log t \rceil} \|\mathbf{v}_{\tilde{\theta}}(k)\|_2\right)^2 := (E(t))^2.$$

Note each $\|\mathbf{v}_{\tilde{\theta}}(k)\|_2$ is an exponential random variable with parameter $\varepsilon$ (mean $\varepsilon^{-1}$). And $E(t)$ is a summation of i.i.d. exponential random variables. Therefore from Theorem 5.1 of [12] we know

$$\Pr[|E(t)| \geq \lambda'/\varepsilon \cdot (\log t + 1)] \leq e^{1-\lambda'}.$$

Take $\lambda' := 4(\log t + 1)$, we know

$$\Pr[|E(t)| \geq 4(\log t + 1)^2 \log^3 T \leq O(1/t^2).$$

Denote by $\lambda(t) := 4(\log t + 1)^2 \cdot \log^3 T$. Then the total noise added up to time $t$ is bounded as follows:

$$\lambda^2(t) \cdot \sum_{n=1}^{n_i(t)} \frac{1}{n^2} = O(\lambda^2(t)).$$

Since $O(\|\mathbf{E}^T(t)\|_2)$ is on a much smaller order, the average error bounds as: $O\left(\frac{\lambda^2(t)}{n_i(t)}\right) = O\left(\frac{\log^4 t \cdot \log^6 T}{n_i(t)}\right)$.

**To summarize the total error induced is at the order of**

$$O\left(\frac{1}{\sqrt{n_i(t)}} + \frac{\log^4 t \cdot \log^6 T}{n_i(t)}\right)$$

The rest of the proof is then similar to the reasoning in the proofs in Section 8.15, we need to bound the following term

$$\Pr\left[S_i^2(t) - \mathbb{E}[S_i^2(t)] \geq \frac{\sqrt{1+\delta_U}-1}{8} c \frac{\log^3 t \log^3 T}{\sqrt{n_i(t)}} - O\left(\frac{1}{\sqrt{n_i(t)}}\right) - O\left(\frac{\log^4 t \cdot \log^6 T}{n_i(t)}\right)\right].$$

After applying the Hoeffding bound, the exponent term is proportional to :

$$-\left(\frac{\sqrt{1+\delta_U}-1}{8}c\frac{\log^3 t \log^3 T}{\sqrt{n_i(t)}} - O\left(\frac{1}{\sqrt{n_i(t)}}\right) - O\left(\frac{\log^4 t \cdot \log^6 T}{n_i(t)}\right)\right)^2 n_i(t).$$

Expand it we will have the positive components coming from the inter-product term: first consider the inter-product term

$$\frac{\sqrt{1+\delta_U}-1}{8}c\frac{\log^3 t \log^3 T}{\sqrt{n_i(t)}} \cdot O\left(\frac{1}{\sqrt{n_i(t)}}\right) \cdot n_i(t)$$
$$= O\left(\log^3 t \cdot \log^3 T\right) < O\left((\log^3 t \cdot \log^3 T)^2\right),$$

For the other inter-product term:

$$\frac{\sqrt{1+\delta_U}-1}{8}c\frac{\log^3 t \log^3 T}{\sqrt{n_i(t)}} \cdot O\left(\frac{\log^4 t \cdot \log^6 T}{n_i(t)}\right) \cdot n_i(t)$$
$$= O\left(\frac{\sqrt{1+\delta_U}-1}{8}c\frac{\log^3 t \log^3 T}{\sqrt{n_i(t)}} \cdot (\log^4 t \cdot \log^6 T)\right)$$
$$\leq O\left(\log^4 t \cdot \log^6 T\right)$$
$$< O\left((\log^3 t \cdot \log^3 T)^2\right),$$

where the first inequality is due to the fact that after we change the bias term we can proved $\sqrt{n_i(t)} = \Omega(\log^3 t \log^3 T)$ a.s., when $t$ is large.

The inner-product term (lower bounded by the first inner-product term):

$$O\left(\left(\frac{\sqrt{1+\delta_U}-1}{8}c\frac{\log^3 t \log^3 T}{\sqrt{n_i(t)}}\right)^2 \cdot n_i(t)\right) = O\left((\log^3 t \cdot \log^3 T)^2\right).$$

Therefore the inter-products (positive exponents) is on a smaller order compared to inner-product terms (negative exponents), and thus can be ignored. We can similarly prove the convergence results.

Meanwhile with changing the bias term from $\sqrt{\log t/n_i(t)}$ to $\frac{\log^3 t \log^3 T}{\sqrt{n_i(t)}}$, workers have stronger incentives to deviate. The difference we need to bound lies in changing the bounding of the following events (in Lemma 8)

$$2c\sqrt{\frac{\log t}{n_i(t)}} + \tau(t) + \frac{C_1}{(\sum_{k\neq i} n_k(t))^2} + \frac{C_1}{(\sum_{k\neq j} n_k(t))^2} < b\underline{L}\Delta ,$$

to the following one

$$2c\frac{\log^3 t \log^3 T}{\sqrt{n_i(t)}} + \tau(t) + \frac{C_1}{(\sum_{k\neq i} n_k(t))^2} + \frac{C_1}{(\sum_{k\neq j} n_k(t))^2} < b\underline{L}\Delta ,$$

from which we know the number of selection after deviating by $\Delta$ is bounded as follows

$$\mathbb{E}[n_i(t)] \leq O\left(\frac{(\log^3 t \cdot \log^3 T)^2}{\Delta^2}\right).$$

Let $t = T$, and when

$$\Delta > O\left(\sqrt{\frac{(\log^3 T \cdot \log^3 T)^2}{T}}\right) = \frac{\log^6 T}{\sqrt{T}} ,$$

we will have $\mathbb{E}[n_i(t)] = o(T)$, from where we can prove a contradiction on non-profitable deviation. $\square$