[Reviews · NeurIPS 2016]

Reviewer 1

Summary

The paper solves a crowd-sourcing mechanism design problem as follows: Given samples x_1, x_2, ... \in R^d, assign these samples to a number of agents to label, so that we can solve a linear regression problem based on these labels. Each agent incurs a cost for improving the accuracy of his labels, and will be compensated by the mechanism. The goal is to induce good behavior in an equilibrium. The result is a UCB-style mechanism for selecting the agents that get the labeling task. The payments, unlike previous work that often use proper scoring rules, is not dependent on the label. Instead, the accuracy of the labels affects the chance that the agent will be selected for the task in the future rounds, and that's how the agents are incentivized to exert the requested level of effort.

Qualitative Assessment

I think this is a very interesting and technically challenging problem, and the solution proposed here is novel and highly non-trivial. It's also interesting to see how the idea of UCB is used for a problem that (at least as far as I can see) is not reducible to MAB. The reason I've taken off a few points is that the presentation not entirely clear. Especially for a complex model like this the authors should probably have spent more effort into clarifying the model, going through simple special cases, and explaining the intuitions behind the proofs. One thing that I found particularly difficult to understand is how the model moves from a static one-shot problem (what's explained at the beginning of Sec 3) to an iterative model where the learner selects a number of agents in each round and asks them to label some of the samples. It appears that the original problem that we want to solve is the one-shot problem, and it is changed to a multi-round problem only to provide incentives for the agents. If that's the case, what does it really mean to have T go to infinity? Also, how is the set of samples in each round selected. Finally, it is not clear to me that this problem does not have a simpler solution. For example, how about simply assigning the tasks to the agents optimally, but assigning a small random selection of tasks to multiple agents, in order to test if they are giving answers at the correct level of accuracy, ensuring this using a proper scoring rule? Minor comments: * learner in several places is misspelled "leaner".

Confidence in this Review

2-Confident (read it all; understood it all reasonably well)


Reviewer 2

Summary

The authors analyze a model of online regression with strategic data sources. A sequence of training feature vectors are drawn from the uniform distribution on the unit ball in R^d. At each iteration the player picks from among a pool of workers to label each of the training samples. Each worker can provide a label with an accuracy that depends on the effort he exerts and the effort is chosen strategically by the player (incurring a linear loss and gaining the payment from the learner). The authors show that a squared loss objective, less the total payment can be optimized at an approximate Bayes-Nash equilibrium. They accomplish this with a very simple form of payment where each worker is paid some target effort level + a premium gamma which vanishes to zero as the number of iterations goes to infinity. Then each worker at each iteration is picked only if his "score" is within a confidence region from the maximum score, where the score depends on the accuracy of his report. This way all workers are incentivized to exert the target effort so that they are selected at every iteration (at least with high probability as time goes by).

Qualitative Assessment

I think the paper is technically strong and addresses an interesting problem in the area of learning with strategic data sources. However, I have some complaints on the formulation and technical part of the results: 1) First of all their notion of \epsilon equilibrium assumes that the players care about average utility and not aggregate! The real incentives for a player to deviate are actually growing with T and so I don't see why players will not deviate for such a huge gain! Especially, when most of the results presented are meaningful in a large T regime. 2) In their deviation analysis the authors seem to neglect the fact that the chosen effort level of the player can be dependent on the label x_i(t) that he is asked to label. Their analysis treats only the case of a uniform deviation to a lower effort level and not a label dependent deviation at each time-step. For instance, in this case, the expected score of two players conditional on the labels they received, will not be the same, creating problems in one crucial step of the analysis. 3) Why is the learner constraint in this simple form of payment? Shouldn't the payment structure be part of the objective? I agree that as T->infinity, the payment structure is optimal as all the welfare is collected. Similarly, why isn't x_i(t) part of the objective. In principle, I could choose which training data to present to each worker and not have it drawn from a distribution. 4) Last, if one ignores the online nature of the formulation here, why isn't the offline problem solved via the techniques of Cai et al., with a more complicated payment structure of course. I think the authors should clarify more the connection with Cai et al.

Confidence in this Review

2-Confident (read it all; understood it all reasonably well)


Reviewer 3

Summary

The author apply the MAB framework to the crowd sourcing. The idea is to maintain an index for each worker. The indices are calculated according to Brier scoring rule and upper confidence bound. The payment rule and the index ensure an \pi-BNE with $\pi$ converge to $0$ if for infinite large $T$. Furthermore, the author shows that the index can be maintained in the online manner and extends the policy to protect the privacy of the workers.

Qualitative Assessment

We are not familiar with the topic/related area discussed in this paper. Therefore, our judgement on the impact and novelty may not be accurate. After quickly going through the related work, we assign 3 to novelty. For the impact, we assign 2 because of the following questions: how could the workers find their own optimal strategy (Is $e^*$ the dominate strategy? In practice, how could the workers find e^* so that they can set e_i to e^*?), so that the NE can be ensured? We know that setting e_i to e^* ensures a BNE, but is the BNE unique? Is there another BNE with much large $pi$? Furthermore, an experiment might be quite helpful to enhance the impact of the research. We could not go through all details in the supplementary file due to the writing style (we stopped at 7.5). The models and theorems introduced in the main paper are intuitive and reasonable. Therefore, we assign 3 to the technical quality. We have to point out that the writing style made us quite confused. First of all, writing a paper without conclusion section is not a good idea(at least to us). Secondly, neither the proof environment nor the QED symbol is used in the proofs, which makes the paper quite hard to read. There are many symbols used in this paper. It would save us a lot time if the author had introduced/summarized the usage before section 3, or provided a short review of these symbols in the proofs, e.g. add short phrase like "let Z be ...." before lemma 7.1 in line 389. Finally, we would suggest \mathbb{R} instead of R for real space and \math{E} instead of E for expectation. For the reasons above, we decided to assign 2 to the clarity and presentation. Some minor errors, which can be easily fixed: In (7.1), "2" missing in the second term of the third line and second and third term of the fourth line. Should $e_i$ be $e$ in the third term in the fourth line of (7.1)?

Confidence in this Review

1-Less confident (might not have understood significant parts)


Reviewer 4

Summary

This paper studies how to incentivize (self-interested) "arms" to exert effort to improve the quality of their predictions in a MAB setting. The setting assumes that no "ground truth" costs are ever revealed, giving the problem an additional technical complication of how to measure the quality of the arms' predictions. They show in certain settings that there exist payment schemes and learning algorithms which have approximate BNE with arms exerting enough effort to give the algorithms nontrivial regret guarantees. They also show how to implement the updates of such algorithms in computationally less expensive way, and also in ways which preserve privacy of the arms.

Qualitative Assessment

The question studied in the paper is interesting, and borrowing the idea from peer prediction to use the other arms' predictions as an unbiased estimator of the quality of one arm's prediction is a nice idea (in particular, because those arms need to be incentivized enough to make reasonably accurate predictions). However, the paper focuses too much on presenting "bells and whistles" rather than giving a deeper understanding of the basic (and main) results. Perhaps reorganizing the paper to only briefly mention the computational/privacy aware variants and giving both more intuition and technical content describing the main result (namely, that there exist \alpha-BNE with small regret) would focus the paper and give the reader a cleaner message of what the paper is doing. This tact would have the added benefit that the reader might be able to better assess the "quantitative" consequences of this work, in that it would leave more room for the authors to ruminate on how much better or worse these bounds are than what one could get in the non-strategic setting, or in various trivial simplifications/special cases of this model. As the paper stands, this reviewer finds it difficult to assess from the main body of the paper alone the technical contribution of the paper (and whether the results follow from a mild reworking of standard proofs or need substantial, new ideas). It is also difficult to assess a theory paper which gives not even a sketch or an outline of a proof in the main body of the paper. It would also be interesting to have some idea of what reasoning goes into designing the payment rules, or some lower bounds, or some impossibility results showing what sorts of BNE cannot be ruled out by any (reasonable) payment rule; since this is not a "PoA" style result that shows *all* apx-BNE have these regret guarantees, only that some will. On a smaller scale, the paper has many typos that a spellchecker would catch, and many grammatical errors which make reading the paper cumbersome. This feels like a rough draft that a few days of love would substantially improve in this regard. It also seems somewhat strange that the assumptions about various parameters are scattered throughout the body of the paper; they should all be in one place or at least in the theorem statement(s). Odds-n-ends - run a spellcheck -"nowadays"? - "degrade the machine learning system's" -> "degrade machine learning systems' " - add "many" before "stable"? - "leveraged" should not be followed by "on" - "indexes" should be "indices" - "computationally light" is a weird phrase. - "the provided incentive much". What? Perhaps "without substantially degrading the incentives guarantee" - "beings" -> "being" - "after receiving the task" should be "after receiving a task" - worth mentioning that different "effort" functions could correspond to different levels of skill in workers - worth mentioning that the per-round payment scheme is "individually rational" - plurality seems to be a regular issue, as well as missing/incorrect articles (the, an, a, etc)

Confidence in this Review

2-Confident (read it all; understood it all reasonably well)


Reviewer 5

Summary

This paper studies a complex setting of active collection of human labels for learning of a regression model of a given form. At each moment, we assign some examples to some workers and define a desired level of worker effort for each example. Our objective is to maximize the difference between the quality of a regression model trained on all the labels after they are collected and the cost of collecting these labels. The critical drawback of the paper is a dramatic lack of clarity. Unfortunately, I recommend to reject it.

Qualitative Assessment

The critical drawback of the paper is a dramatic lack of clarity. The following facts could be inferred from the text, but are not explicitly and clearly stated in appropriate places: 1) From line 95, I conclude that the set X of unlabeled training examples (feature vectors) is sampled before the learner starts its procedure and thus is known for the learner. 2) From lines 101-102, I conclude that each example is assigned to exactly one worker. 3) From lines 122-123, I conclude that ${\sigma_i(\cdot)}_{i=1,…,N}$ are known for the learner. [Additionally: when assuming that ${\sigma_i(\cdot)}_{i=1,…,N}$ are unknown for the learner, I have the following questions: a. The order of samples may influence our choice of workers at different steps (since it influences the quality of the model at a particular step t and, thus, indices $S_i(t)$). How is this order defined? b. Index $S_i(t)$ depends on the effort levels we chose for the worker $i$ in the past. If we had chosen different effort levels for different workers in the past, comparison of these workers at step t is biased. Is not it a problem?] Contradictions in formulations: 1) According to my conclusion 3, the observed labels $\tilde{y}_i$ do not bring any additional information about workers, and I do not see any motivation to choose workers on the basis of these observations, in particular, on the basis of S_i(t) (I also failed to find any motivation for it in the text, e.g., in Section 3.1, Assignment). 2) The optimization problem in equation after line 114 does not imply any choice of workers. How is this problem related to ones considered in Section 4? 3) Lines 192-194: first authors claim that assigning T samples to each worker is only ideal (it is unclear why and whether this “better” is formalized or is justified by some reasons beyond the problem formalization), but not necessary; then they formulate an optimization problem in the equation after line 194 where this condition is predefined. So, do we accept this condition as an assumption? Finally (and most importantly, since this does not allow me to understand the main statements, e.g., Theorem 4.2), I do not understand how a worker chooses an effort level for each task. Namely, how does he estimate an “expected payment” mentioned in line 107? How does our desired effort level (chosen by the learner for this example) can influence his choice? Related to the above question: motivation for assigning examples to a suboptimal worker in lines 208-211 is not formalized. Do we assume that our previous payments to other workers influence an estimate of “expected payment” by a given worker? Some other questions and corrections: Line 152: What does “as with an index policy” relate to? Line 153: “benefits” -> “benefit” Lines 163-164: “The reason we adopt such a notion is in a sequential setting it is generally hard to achieve strict BNE or even other stronger notion” – unclear construction Line 211: “a effort level” -> “an effort level” What is “e” in the equation after line 212? Unfortunately, poor presentation of the paper did not allow me to understand the problem formulation and, as a result, statements presented in this paper. I recommend to reject it.

Confidence in this Review

1-Less confident (might not have understood significant parts)


Reviewer 6

Summary

The paper introduces a new online algorithm for dynamic data acquisition under bandit setting called SR-UCB, and its variant, PSR-UCB that can preserve (O((1/\log T), O(1/T^2)) differential privacy. Authors also provide a theoretical analysis that algorithm achieves O(\sqrt{\log T / T})-Bayesian Nash Equilibrium.

Qualitative Assessment

The proposed novel SR-UCB framework collects data from strategic data sources to train regression model. Motivated by MAB framework, this paper extends strategic learning to a UCB based index rule for worker selection, to acquire high-quality data in the long run. The paper mainly focus on training linear regression model while also discuss about extending to non-linear case. The contribution of the proposed framework is sound and discussion is thorough. The paper is well written. The sketch of the proof is clear and seems solid (however I did not check all proofs in detail).

Confidence in this Review

1-Less confident (might not have understood significant parts)